



**Constraining N$_2$O emissions since 1940 using firn air isotope measurements in**
**both hemispheres**
**M. Prokopiou[1], P. Martinerie[2], C. J. Sapart[1,3], E. Witrant[4], G. A. Monteil[1,5], K.**
**Ishijima[6], S. Bernard[2], J. Kaiser[7], I. Levin[8], T. Sowers[9], T. Blunier[10], D.**
**Etheridge[11], E. Dlugokencky[12], R. S. W. van de Wal[1], T. Röckmann[1]**
[1] Institute for Marine and Atmospheric research Utrecht, Utrecht, The Netherlands
[2] University of Grenoble Alpes/CNRS, LGGE, F-38000 Grenoble, France
[3] Laboratoire de Glaciologie, ULB, Brussels, Belgium
[4] University of Grenoble Alpes/CNRS, GIPSA-Lab, F-38000 Grenoble, France
[5] Department of Physical Geography and Ecosystem Science, Lund University, Lund, Sweden
[6] National Institute of Polar Research, Tokyo, Japan
[7] Centre for Ocean and Atmospheric Sciences, School of Environmental Sciences, University
of East Anglia, Norwich, United Kingdom
[8] Institute of Environmental Physics, Heidelberg University, Germany
[9] Earth and Environmental Systems Institute, Pennsylvania, USA
[10] Centre for Ice and Climate, Niels Bohr Institute, Copenhagen, Denmark
[11] CSIRO Marine and Atmospheric Research, Victoria, Australia
[12] NOAA Earth System Research Laboratory, Boulder, Colorado, USA
**Abstract**
N$_2$O is currently the 3$^{rd}$ most important anthropogenic greenhouse gas in terms of radiative
forcing and its atmospheric mole fraction is rising steadily. To quantify the growth rate and its
causes, we performed a multi-site reconstruction of the atmospheric N$_2$O mole fraction and
isotopic composition using firn air data collected from Greenland and Antarctica in
combination with a firn diffusion and densification model. The multi-site reconstruction
showed that while the global mean N$_2$O mole fraction increased from $(290\pm1)$ nmol mol$^{-1}$ in
1940 to $(322\pm1)$ nmol mol$^{-1}$ in 2008 the isotopic delta [values] of atmospheric N$_2$O decreased
by $(-2.2\pm0.2)$ ‰ for $\delta^{15}N^{av}$, $(-1.0\pm0.3)$ ‰ for $\delta^{18}O$, $(-1.3\pm0.6)$ ‰ for $\delta^{15}N^{\alpha}$, and $(-2.8\pm0.6)$
‰ for $\delta^{15}N^{\beta}$ over the same period. The detailed temporal evolution of the mole fraction and



isotopic composition derived from the firn air model was then used in a two-box atmospheric
model (comprising a stratospheric and a tropospheric box) to infer changes in the isotopic
source signature over time. The precise value of the source strength depends on the choice of
the $N_2O$ lifetime, which we choose to be $123^{+29}_{-19}$ a. Adopting this lifetime results in total
average source isotopic signatures of $(-7.6\pm0.8)$ ‰ (vs. Air-$N_2$) for $\delta^{15}N^{av}$, $(32.2\pm0.2)$ ‰ (vs.
VSMOW) for $\delta^{18}O$, $(-3.0\pm1.9)$ ‰ (vs. Air-$N_2$) for $\delta^{15}N^{\alpha}$, and $(-11.7\pm2.3)$ ‰ (vs. Air-$N_2$) for
$\delta^{15}N^{\beta}$ over the investigated period. $\delta^{15}N^{av}$ and $\delta^{15}N^{\beta}$ show some temporal variability while the
other source isotopic signatures remain unchanged. The $^{15}N$ site-preference $(= \delta^{15}N^{\alpha} - \delta^{15}N^{\beta})$
can be used to reveal further information on the source emission origins. Based on the
changes in the isotopes we conclude that the main contribution to $N_2O$ changes in the
atmosphere since 1940 is from soils, with agricultural soils being the principal anthropogenic
component which is in line with previous studies.

## 1   Introduction

The rise of nitrous oxide ($N_2O$) since pre-industrial times contributes significantly to radiative
forcing (Forster et al., 2007). Over the past four decades, the $N_2O$ mole fraction has increased
by 0.25 % per year, reaching 324 nmol mol$^{-1}$ in 2011 (IPCC, ch.6, 2013). Therefore, the
understanding of the biogeochemical cycle of $N_2O$ is important for a reliable assessment of
future climate change. In addition, the destruction of $N_2O$ in the stratosphere provides an
important source of nitrogen oxides ($NO_x$), which contribute to stratospheric ozone depletion
(Ravishankara et al., 2009).
Natural sources of $N_2O$ are microbial processes in soils and oceans, which produce $N_2O$
during nitrification and denitrification (Bouwman et al., 2013; Loescher et al., 2012; Santoro
et al., 2011; Galloway et al., 2004; Pérez et al., 2001; Yung and Miller, 1997; Kim and Craig,
1993). The increase of $N_2O$ since pre-industrial times (hereafter referred to as
"anthropogenic" increase) has been attributed largely to increased microbial production,
resulting from the increased use of nitrogen fertilizers in agriculture. Industry (especially
nylon production) and fossil fuel combustion present a smaller contribution to the
anthropogenic source (Davidson, 2009; Kroeze et al., 1999; Mosier et al., 1998). $N_2O$ is
primarily destroyed in the stratosphere via UV photolysis (90%) and reactions with excited
oxygen atoms (10 %) (Minschwaner et al., 1993), with a minor $N_2O$ fraction removed by
surface sinks (Syakila, 2010).





Estimates of the total $N_2O$ source strength from various bottom-up and top-down studies
suggest a mean value of roughly 17 Tg $a^{-1}$ N equivalents at present. However, the range in
both approaches is large, especially for bottom-up estimates, which range between 8.5 and
27.7 Tg $a^{-1}$ N, whereas top-down estimates range between 15.8 and 18.4 Tg $a^{-1}$ N (Potter et
al., 2011 and references therein). Besides the total source strength, the contributions of
individual source processes are also poorly constrained. Due to the long steady-state lifetime
of $N_2O$ in the atmosphere ($123^{+29}_{-19}$ a; SPARC Lifetimes Report 2013), temporal and spatial
gradients are small, making it difficult to resolve localised sources.
Measurements of the isotopic composition of $N_2O$ may help to constrain the atmospheric $N_2O$
budget. The $N_2O$ molecule is linear (NNO) and the two N atoms are chemically
distinguishable; thus they tend to attain different isotopic compositions. Beyond oxygen
($\delta^{18}O$, $\delta^{17}O$) and average $\delta^{15}N^{av}$ ("bulk") signatures, $N_2O$ also displays site specific $^{15}N$
isotopic information. Site preference ($\delta^{15}N^{sp}$) is defined as the difference in $\delta^{15}N$ between the
central (2, $\mu$ or $\alpha$) and terminal position (1, $\tau$ or $\beta$) of N atoms in $N_2O$ (Kaiser, 2002;
Brenninkmeijer and Röckmann, 2000; Yoshida and Toyoda, 1999),
i.e. $\delta^{15}N^{sp} = \delta^{15}N^{\alpha} - \delta^{15}N^{\beta}$. For consistency with many recent publications in the field, we here
adopt the nomenclature from Yoshida and Toyoda (1999), $\alpha$ and $\beta$, for the two positions.
The different sources and sinks of $N_2O$ are associated with characteristic fractionation
processes leading to different isotope ratios. For example, microbial sources emit $N_2O$ that is
depleted in $^{15}N$ and $^{18}O$ relative to the tropospheric background. $N_2O$ that returns from the
stratosphere after partial photochemical removal is enriched in both heavy isotopes (Yoshida
and Toyoda, 2000; Yung and Miller, 1997; Kim & Craig, 1993). Stratospheric $N_2O$ also has a
high $^{15}N$ site-preference compared to tropospheric $N_2O$. The observed enrichment is caused
by kinetic isotope fractionation in the stratospheric sink reactions (Kaiser et al., 2006; 2002;
Park et al., 2004; Röckmann et al., 2001; Yoshida and Toyoda, 2000).
The multi-isotope signature of $N_2O$ adds useful constraints on its budget. In particular, when
the isotopic composition of tropospheric $N_2O$ is combined with the fractionation during its
removal in the stratosphere, the isotopic composition of the global average source can be
determined (Ishijima et al., 2007; Bernard et al., 2006; Röckmann et al., 2003; Kim and Craig,

89    1993).

The temporal variations of the $N_2O$ isotopic composition are difficult to quantify on a short
timescale because of its long residence time in the atmosphere. Longer time scales can be





reconstructed by using air trapped in Arctic and Antarctic firn and ice which provides a
natural archive of past atmospheric composition. The firn phase is the intermediate stage
between snow and glacial ice, which constitutes the upper 40-120 m of the accumulation zone
of ice sheets. Within the firn, air exchanges relatively freely in the upper layers and with the
overlying atmosphere (convective zone). With increasing depth the air pores shrink in size
due to firn compaction, and air mixes primarily via slow diffusion in the diffusive zone. At
densities larger than $\approx 815$ kg m$^{-3}$, air is permanently trapped in closed bubbles in the ice and
totally isolated from the atmosphere. The precise age range of air that can be retrieved from
polar firn between the surface and bubble close-off depends on site specific characteristics
like temperature, accumulation rate and porosity and typically ranges from several decades to
120 years.
For $N_2O$, a number of studies have reported isotope measurements from different Arctic and
Antarctic firn drilling sites showing a steady decrease of the heavy isotope content of $N_2O$
over the past decades (Park et al., 2012; Ishijima et al., 2007; Bernard et al., 2006; Röckmann
et al., 2003; Sowers et al., 2002). A more recent study by Park et al. (2012) attempted to
reconstruct the long-term trends in $N_2O$ isotopic compositions and its seasonal cycles to
further distinguish between the influence of the stratospheric sink and the oceanic source at
Cape Grim, Tasmania, demonstrating that isotope measurements can help in the attribution
and quantification of surface sources in general.
Taking into account the long atmospheric lifetime of $N_2O$ and the fact that both hemispheres
are well mixed on annual timescales, it is reasonable to assume that the results from these
studies are representative for the global scale. However care needs to be taken because small
differences in the diffusivity profiles of the firn column lead to large effect on the isotope
signature (Buizert et al. 2012). Interestingly, for atmospheric methane ($CH_4$), another
important greenhouse gas, a recent multi-site analysis on the carbon isotopic composition of
showed large differences among reconstructions from different sites (Sapart et al., 2013). In
particular, firn fractionation effects related to diffusion and gravitational separation and their
implementation in models (Buizert et al., 2012) have large effects on the reconstructed
signals. Small differences in the diffusivity profiles of the firn column lead to large effects on
the isotope signatures. Therefore, more robust results may be obtained by combining isotope
information from a number of different sites in a multi-site reconstruction, including a critical
evaluation of diffusivity profiles.



Here we combine new $N_2O$ isotope measurements from the NEEM site in Greenland with
previously published firn air $N_2O$ isotope records from 4 different sites from Greenland and
Antarctica to reconstruct records of the $N_2O$ isotopic composition over the last 70 years. We
use the multi-gas firn transport model established by the Laboratoire de Glaciologie et
Géophysique de l'Environnement and Grenoble Image Parole Signal Automatique (LGGE-
GIPSA) to obtain an atmospheric scenario that is constrained by and consistent with all
individual sites (Allin et al., 2015; Witrant et al., 2012; Wang et al., 2012; Rommelaere et al.,
1997). We then use an isotope mass balance model to infer the changes in the isotopic
signature of the $N_2O$ source over time to investigate possible changes in the source mix.
**2    Materials and Methods**
**2.1    Firn air Sampling**
New firn air samples added in this study to the total dataset were collected in 2008 and 2009
during the firn campaign (Buizert et al., 2011) as part of the North Eemian Ice Drilling
programme (NEEM) in Greenland (77.45º N 51.06º W). These data are combined with
existing firn air data from four other sites. Information on the locations is provided in Table 1.
The firn air collection procedure is described in detail by Schwander et al. (1993). Here a
brief description is presented. Essentially a borehole is drilled in the firn to a certain depth and
then the firn air sampling device is inserted into the borehole. The device consists of a
bladder, a purge line and a sample line. When the sampling device reaches the desired depth
the bladder is inflated to seal the firn hole and isolate the air below the bladder from the
overlying atmosphere, and air is pumped out from the pore space below the bladder.
Continuous online $CO_2$ concentration measurements are performed to verify that no
contamination with contemporary air occurs during the extraction procedure. After the
contaminating air has been pumped away, firn air is collected in stainless steel, glass or
aluminium containers.
**2.2    $N_2O$ isotope analysis**
The firn air samples from NEEM are analyzed for $N_2O$ isotopocules at the Institute for
Marine and Atmospheric research Utrecht (IMAU). The $N_2O$ mole fraction and isotopic
composition are measured using continuous flow isotope ratio mass spectrometry (IRMS).
The method is described in detail by Röckmann et al. (2003b). Here only a brief summary is




given. The firn air sample (333 mL) is introduced into the analytical system at a flow rate of
50 mL/min for 400 s. After $CO_2$ is removed chemically over Ascarite, $N_2O$ and other
condensable substances are cryogenically preconcentrated. After cryo-focusing the sample the
remaining traces of $CO_2$ and other contaminants are removed on a capillary GC column
(PoraPlot Q, 0.32 mm i.d., 25 m). The column is separated into a pre-column and an
analytical column. This set-up eliminates interferences from other atmospheric compounds
that have much longer retention times. Finally the sample is transferred to the IRMS via an
open split interface. For the new NEEM samples reported here, each firn air sample has been
measured five times. Before and after each sample we measured five aliquots of air from a
reference cylinder with known isotopic composition and mole fraction for calibration
purposes.
$\delta^{15}N$ values are reported with respect to Air-$N_2$ while $\delta^{18}O$ refers to Vienna Standard Mean
Ocean Water (VSMOW). As laboratory reference gas we used an atmospheric air sample with
an $N_2O$ mole fraction of 318 nmol mol$^{-1}$ and $\delta$ values of (6.4±0.2) ‰ for $\delta^{15}N^{av}$ vs. Air-$N_2$,
(44.9±0.4) ‰ for $\delta^{18}O$ vs. VSMOW. The intramolecular $\delta^{15}N^{av}$ values of the air standard are
$\delta^{15}N^{\alpha}$ = (15.4±1.2) ‰ and $\delta^{15}N^{\beta}$ = (–2.7±1.2) ‰. The calibration of the intramolecular
distribution follows Toyoda and Yoshida (1999). Typically the 1σ standard deviation of
replicate sample measurements are 0.1 ‰ for $\delta^{15}N$, 0.2 ‰ for $\delta^{18}O$ and 0.3 ‰ for $\delta^{15}N^{\alpha}$ and
$\delta^{15}N^{\beta}$.

## 173   2.3   Modelling trace gas transport in firn

In firn air, the interstitial gas is not yet isolated in closed-off bubbles, so diffusion processes
and gravitational separation alter mole fractions and isotope ratios over time. Thus, firn air
measurements cannot be used directly to derive the atmospheric history of trace gas
signatures. Over time, atmospheric compositional changes are propagated downwards into the
firn based on the diffusivity of the atmospheric constituent in question. Firn air diffusion
models take these effects into account and thereby allow reconstruction of changes in the
atmospheric composition from the firn profile.
In this study we use the LGGE-GIPSA firn air transport model to reconstruct the temporal
evolution of $N_2O$ mole fraction and isotopic composition from the measured firn profiles
(Allis et al., 2015; Witrant et al., 2012; Wang et al., 2012; Rommelaere et al., 1997).



In the "forward version" of LGGE-GIPSA, a physical transport model uses a historic
evolution of atmospheric $N_2O$ mole fractions to calculate the vertical profiles of mole
fractions in firn. For the isotopocules, further simulations are performed separately to
calculate their respective vertical profiles. Important parameters needed to constrain the
model are the site temperature, accumulation rate, depth of the convective layer and close-off
depth, together with profiles of firn density and effective diffusivity. The latter parameter is
determined as a function of depth for each firn-drilling site by modelling the mole fractions in
firn for trace gases with well known atmospheric histories (Buizert et al., 2012; Witrant et al.,
2012; Rommelaere et al., 1997; Trudinger et al., 1997). A multi-gas constrained inverse
method (Witrant et al., 2012) is used to calculate the effective diffusivity of each site for each
specific gas. It is noteworthy that diffusivity is not constrained equally well at all sites
because different sets of constraints (e.g. number of available reference gases) are used at
different sites and because of different depth resolutions.
A Green-function approach as presented by Rommelaere et al. (1997), with an extension for
isotopic ratios and revised to take into account the sparsity of the measurements (Witrant and
Martinerie, 2013; Martinerie et al., 2012; Wang et al., 2011) is used to assign a mean age and
age distribution to a certain depth.
Due to the long $N_2O$ residence time in the atmosphere, the global variability of the isotopic
composition of $N_2O$ is very small and no significant variations between individual
background locations have been detected so far (Kaiser et al., 2003). In particular, the isotope
ratio difference between northern and southern hemisphere tropospheric air is expected to be
only –0.06 ‰ (based on an interhemispheric mole fraction gradient of 1.2 nmol $mol^{-1}$ [Hirsch
et al. 2006] and isotope ratio difference of –15 ‰ between average source and average
tropospheric isotope delta). These differences are within the uncertainties of the firn air
measurements used here and therefore the data from the northern and southern hemisphere are
combined into a single dataset without including an interhemispheric gradient.
With the multi-site reconstruction method, we used the measurements from six firn air
drillings at five sites (NEEM-09, NEEM-EU-08, NGRIP-01, BKN-03, DC-99, DML-98) to
constrain our model and determine a set of atmospheric reconstructions that fits all sites. Data
from Ishijima et al. (2007) and Sowers et al. (2002) [NGRIP-01 and SP-01, SP-95
respectively] were not included in our multi-site reconstruction because no data for $\delta^{15}N^{\alpha}$ and





$\delta^{15}N^\beta$ were published for those sites. These datasets were used for independent validation of
$\delta^{15}N^{av}$ and $\delta^{18}O$.
To quantify the isotope fractionation due to diffusion and gravitational settling within the firn,
a forward firn transport model simulation was carried out with a realistic $N_2O$ mole fraction
scenario (based on the Law Dome record, MacFarling Meure et al., 2001), but with a constant
isotopic $N_2O$ history. This allows determining the role of transport isotope fractionation
occurring in the firn, in the absence of isotopic changes in the atmosphere. The results are
used to subtract the firn fractionation effects from the measured signals, which can then be
used to assess the atmospheric history. Compared to the signal, the effect of firn fractionation
is minor for $\delta^{15}N$, but important for $\delta^{18}O$ especially at the lower accumulation rates in the
Southern Hemisphere (see Appendix A).
The deepest firn data from each site provide constraints furthest back in time and the oldest
air samples that are included in the inversion are from the DML-98 and DC-99, which extend
the reconstruction of atmospheric $N_2O$ back to the early 20[th] century (Röckmann et al., 2003).
At the same, the correction for isotopic fractionation in firn is most uncertain for the deepest
samples, where strong differences between individual firn air models have been reported
(Buizert et al., 2012).

## 2.4   Scaling of different data sets

At present, no international reference materials for the isotopic composition of $N_2O$ exist.
Kaiser et al. (2003) and Toyoda et al. (1999) linked the isotopic composition of $N_2O$ in
tropospheric air to the international isotopes scales for nitrogen isotopes (Air-$N_2$) and oxygen
isotopes (either VSMOW or Air-$O_2$). Our measurements are linked to a standard gas cylinder
of tropospheric air with known $N_2O$ mole fraction and isotopic composition based on the
scale of Kaiser et al. (2003) for $\delta^{15}N^{av}$ and $\delta^{18}O$ values and Yoshida and Toyoda (1999) for
position dependent $^{15}N$ values. However, the reference air cylinder used for the calibration
was exhausted and had to be replaced three times over the years in which the different
measurement that we combine in this study were performed. Although the cylinders were
carefully compared, the long-time consistency of the isotope scale could not be guaranteed
because long-time isotope standards are not available. In fact, analysis of the data from the
convective zone for the different sites, show small but significant differences from the
temporal trends that are well established from previously published data from the German



Antarctic Georg von Neumayer station for 1990 to 2002 (Röckmann and Levin; 2005). The
linear trends reported in that paper are ($-0.040\pm0.003$) ‰ $a^{-1}$ for $\delta^{15}N^{av}$, ($0.014\pm0.016$) ‰ $a^{-1}$
for $\delta^{15}N^{\alpha}$, ($-0.064\pm0.016$) ‰ $a^{-1}$ for $\delta^{15}N^{\beta}$ and ($-0.021\pm0.003$) ‰ $a^{-1}$ for $\delta^{18}O$. Since they were
derived from direct air samples (unaffected by firn fractionation), these trends can be used as
a reference to re-scale the different firn air results from different dates. To do so, data from
the diffusive zone ($\rho < 815$ kg $m^{-3}$) for each individual site were scaled to one reference site,
DC-99, taking into account the temporal differences in sampling and the model-assigned
mean age of the firn air samples (see below). DC-99 was chosen as reference site because it
has most measurements in the diffusive zone. Also, the precision of these measurements was
high because high volume cylinders were available from which many measurements could be
performed and averaged. To test the sensitivity to the choice of reference site, we repeated the
re-scaling using NEEM-09 as reference, which generated almost identical results within
uncertainty bars (Appendix C).
The average difference between the samples from the diffusive zone at a given site and the
interpolated DC-99 results was compared to the expected temporal trend between the
sampling date of each station and DC-99, using the temporal trends established by Röckmann
and Levin (2005), as shown in the equations below. The effect of this scaling is that the
temporal trend in the past decade is effectively forced to follow the atmospheric
measurements at Neumayer station (Röckmann and Levin, 2005).
After re-scaling the firn isotopic data we detected some individual data points that clearly
deviated from the general trends. These were considered outliers, because they exceeded the
$2\sigma$ error, and were removed from the dataset. All of these values are site-specific $^{15}N$ values,
specifically, the following, were excluded: NEEM-EU-08 hole depth $-4.9$ m, $-34.72$ m, $-$
61.95 m and $-74.5$ m, and NEEM-09 hole depth 1.0 m, 0.2 m and $-69.4$ m.
The mole fraction data that can be obtained from the NEEM air isotope measurements were
substituted with more precise measurements of $N_2O$ mole fraction by the Commonwealth
Scientific and Industrial Research Organisation (CSIRO) the Institute of Environmental
Physics, University of Heidelberg (IUP), the Centre of Ice and Climate, University of
Copenhagen (CIC) and National Oceanic and Atmospheric Administration (NOAA). In this
way we combine all available $N_2O$ mole fraction data.
The mole fraction data were scaled to the most recent international scale, NOAA-2006A from
the CSIRO scale or the NOAA-2000 scale. Conversion of the NOAA-2000 data to the





NOAA-2006A scale is done using a conversion factor available by National Oceanic and
Atmospheric                 Administration                (NOAA)
(http://www.esrl.noaa.gov/gmd/ccl/scales/N2O_scale.html). Converting from the CSIRO to
the NOAA-2006A scale, though, requires the reference cylinder details, which were not
available. Instead we used a trend scenario, based on the CSIRO atmospheric scale combined
with Law Dome data and assuming a constant interhemispheric gradient. This trend scenario
was then compared with the data provided on NOAA-2006A scale, and a polynomial fit was
generated, which was then used to convert the data to the NOAA-2006A scale. All results
presented in the next section are based on the scaling procedure and removal of the outliers as
described above (Appendix B).

## 288    2.5   Global $N_2O$ (isotope) budget calculations

The tropospheric budget is controlled by $N_2O$ emissions from natural and anthropogenic
sources at the surface and by the exchange between troposphere and stratosphere. A simple
two-box model is used to quantitatively understand the emissions and the budget changes of
$N_2O$. The model consists of a tropospheric $N_2O$ reservoir (index T) into which $N_2O$ is emitted
from natural ($E_{nat}$) and anthropogenic ($E_{anth}$) sources. $N_2O$ is then transported to the
stratosphere (index S) where part of it is destroyed by photochemical reactions ($F_{sink}$), and the
remainder returns from the stratosphere to the troposphere ($TS_{exch}$).
The change in the tropospheric $N_2O$ reservoir is given by the following mass balance
equations (Allin et al, 2015):
$$n_T \frac{d\chi_T}{dt} = E_{nat} + E_{anth} - F_{exch}(\chi_T - \chi_S) \qquad (1)$$
$$n_S \frac{d\chi_S}{dt} = F_{exch}(\chi_T - \chi_S) - L \qquad (2)$$
where $n$ is the amount of air and $\chi_S$ and $\chi_T$ are the mole fractions of $N_2O$ in the stratosphere
and troposphere respectively. Annual fluxes between the two reservoirs, $F_{exch}$, are calculated
based on previous estimates (Appenzeller et al., 1996; Holton et al., 1990). The loss due to
stratospheric sink is determined by:
$$L = \frac{n_T \chi_T + n_S \chi_S}{\tau} \qquad (3)$$
where $\tau$ is the atmospheric lifetime of $123^{+29}_{-19}$ a.





The isotopic budgets are calculated by simply multiplying the reservoir sizes with the
corresponding $\delta$ values of the different flux terms:
$n_T \frac{d\chi_T \delta_T}{dt} = E_{nat}\delta_{nat} + E_{anth}\delta_{anth} + F_{exch}(\chi_S \delta_S - \chi_T \delta_T)$  *(4)*
$n_S \frac{d\chi_S \delta_S}{dt} = F_{exch}(\chi_T \delta_T - \chi_S \delta_S) - L\delta_L$  *(5)*
Solving equations 4 and 5 and substituting equations 1 and 2 we reach the final isotope
equations:
$n_T \frac{d\delta_T}{dt} = \frac{E_{nat}}{\chi_T}(\delta_{nat} - \delta_T) + \frac{E_{anth}}{\chi_T}(\delta_{anth} - \delta_T) + \frac{F_{exch}\chi_S}{\chi_T}(\delta_S - \delta_T)$  *(6)*
$n_S \frac{d\delta_S}{dt} = \frac{F_{exch}\chi_T}{\chi_S}(\delta_T - \delta_S) - \frac{L}{\chi_S}\varepsilon_L$  *(7)*
where $\delta_T$ is either $\delta^{15}N^{av}$, $\delta^{18}O$, $\delta^{15}N^{\alpha}$, $\delta^{15}N^{\beta}$ from the multi-site reconstruction as shown
below. $\delta_{nat}$ and $\delta_{anth}$ is the isotopic composition of the natural and anthropogenic $N_2O$ source,
respectively (our target quantity). $\varepsilon_L$ is the apparent fractionation factor associated with
stratospheric destruction.
$\delta_S$ is also not known in this case, but can be calculated using the analogue from Röckmann et
al. (2003) by employing the observed apparent Rayleigh fractionation in the stratosphere
($\varepsilon_{app}$). Based on this, the relative isotope ratio difference between the stratosphere and the
troposphere can be calculated by:
$\delta_S = \left[(\delta_T + 1)\left(\frac{\chi_S}{\chi_T}\right)^{\varepsilon_{app}} - 1\right]$  *(8)*
where $\varepsilon_{app}$ represents the stratospheric fractionation constant associated with this removal
process. Here, we used the average $\varepsilon_{app}$ of all lowermost stratospheric measurements from
Kaiser et al. (2006) (Table 3). Note that slightly different fractionations $\varepsilon_{app}$ have been used
in previous studies by Röckmann (2001) and Park et al. (2012; 2004) and the sensitivity to
these differences will be examined below.
Furthermore we assume that the $N_2O$ lifetime and $\varepsilon_{app}$ remained constant from pre-industrial
time to 2008, thus the annual strength removal can be scaled down from its current value at
$\chi_T = 322$ nmol mol$^{-1}$ to the pre-industrial level of $\chi_{T,pi} = 270$ nmol mol$^{-1}$ and the relative





enrichment of stratospheric $N_2O$ relative to tropospheric $N_2O$ described by Eq. 8 remains
constant over time. The effect of changing the $N_2O$ lifetime is examined below.
Furthermore it is hypothesized that during the pre-industrial period only natural emissions
occurred without any anthropogenic input. After the industrialization ($\approx$1750) any increase in
the emission budget is considered to be due to anthropogenic input while natural emissions
remain constant, allowing separation of $E_{nat}$ and $E_{anth}$.
Hence, the isotopic signature of the pre-industrial (natural) $N_2O$ source calculated this way
also represents the isotopic signature of the natural source at present, and consequently the
average isotopic composition of $N_2O$ originating from "anthropogenic" sources ($\delta_{anth}$) can be
estimated.
**2.6   Uncertainty estimation using random scenarios**
The precision of the calculated $N_2O$ emissions ($E_{nat}$, $E_{anth}$) depends primarily on the precision
of the atmospheric reconstruction of the $N_2O$ mole fraction ($\chi_T$). However, the uncertainty
envelope provided by the firn air reconstruction is insufficient to quantify the uncertainty on
the atmospheric $N_2O$ reconstruction: the year-to-year variability of $N_2O$ is constrained by the
$N_2O$ lifetime in the troposphere, which is very small in comparison to the width of the
reconstructions confidence interval. Possible realistic $N_2O$ scenarios are scenarios that are
within the confidence intervals provided by the atmospheric reconstructions, and that have
realistic year-to-year variability.
Mathematically, this can be represented by an uncertainty variance covariance matrix **B**,
where the diagonal elements (variances) are the yearly uncertainties on the atmospheric $N_2O$
mole fractions, and the off-diagonals are the covariances of the uncertainties of different
years. The covariance between the uncertainty on the reconstruction in one year $i$ and the
uncertainty in another year $j$ is defined as:
$cov(i,j) = r_{i,j}\sigma_i\sigma_j$                                                    (9)
$r_{i,j} = f(|i\text{-}j|)$                                                              (10)
The correlation ($r_{i,j}$) is maximum between two consecutive years, and decreases as the time
difference increases.




We generated an ensemble of 50 random realistic $N_2O$ scenarios within the uncertainty
envelope of the firn atmospheric $N_2O$ reconstruction constrained by the covariance matrix **B**.
For each of these atmospheric $N_2O$ scenarios, we calculated the corresponding $N_2O$ emission
time series. The range of emissions from these scenarios then provides a realistic estimate for
the uncertainty in $N_2O$ emissions.
We carried out the same analysis for the different $N_2O$ isotopocules: for each isotopocule ($\delta$
value), we generated a covariance matrix $\mathbf{B}^{\delta}$, constrained by the uncertainty ranges provided
by the atmospheric reconstructions and the correlation coefficients defined in Eq.6 and Eq.7
to generate a set of 50 random scenarios within the uncertainty envelopes. For each of these
random scenarios, we calculated the corresponding source signature scenario and the range in
the results provides an uncertainty estimate of the isotopic source signatures.
**3    Results**
**3.1    Mean age**
The mean age of $N_2O$ in air sampled from different depths in the firn for all datasets that are
used in this study is shown in Fig. 1. The strong change in the mean age gradient that is
clearly visible in each profile reflects the transition between the diffusive and bubble close-off
zones, which occurs at a specific depth and mean age for each site (marked with x on Fig.1).
Fig. 1 also shows that for each site the few samples that are collected within the bubble close-
off zone provide the constraints for most of the reconstructed record (for instance, at BKN-03,
50 m depth is the beginning of the bubble close-off zone). In addition to the mean age, the
width of the age spectrum also increases with depth. Therefore, the temporal resolution of
signals that can be reconstructed from the firn air measurements reduces with depth and
approaches the one of ice core samples towards the bottom of the bubble close-off zone.
The Greenland sites (NH) have similar meteorological and glaciological conditions (Table 1),
thus the differences between the mean age profiles in Fig. 1 are small. The Antarctic sites
(SH) show clear differences because the meteorological and glaciological variables differ
strongly from site to site. As a result the firn-ice transition is at a different depth for each
location (e.g., the firn-ice transition zone for DML-98 is located at about 73.5 m compared to
about 99.5 m at DC-99).





### 3.2 Experimental results and multi-site reconstruction


Mole fraction and isotopic composition of $N_2O$ in firn air are presented versus depth of the
firn air sampling in the middle panels of Fig. 2. The mole fraction decreases with depth in
qualitative agreement with the known increase of $N_2O$ in the atmosphere over time. In
contrast, all isotope deltas slowly increase with depth in the upper firn and show stronger
heavy isotope enrichment in the close-off zone, both indicating heavy isotope depletion in
atmospheric $N_2O$ with time.
The atmospheric history that has been reconstructed from these firn datasets using the multi-
site inversion (using the data from NEEM-09, NEEM-EU-08, NGRIP-01, BKN-03, DC-99,
DML-98) as described in section 2.4 is shown in the left panels of Fig. 2. The solid line shows
the scenario that leads to the best fit with all firn data in the middle panel, and the dashed lines
show the upper and lower range of possible scenarios that would still produce an acceptable
fit to the data within the uncertainty bars. Color-coded symbols show data plotted at their
respective mean age (as derived from the firn air model). When the best-fit scenario is used as
input for the forward firn air model for each individual site, the model produces the vertical
profiles that are shown as coloured lines together with the data in the middle panels. For the
sites that were included in the multi-site reconstruction, the firn profiles based on the best-fit
scenarios generally match the experimental data points well, which is expected after a
successful inversion procedure and with consistent data sets. The right panels in Fig. 2 show
the differences between these model results and the data. For the data that were used in the
multi-site inversion the model-data differences are generally very small, although individual
firn drilling sites in some cases show small systematic deviations, in particular in the close-off
zone. This means that when inversions would have been performed on individual sites, the
optimal reconstructions would be slightly different. The advantage of the multi-site
reconstruction is that the reconstructed scenario is constrained by all sites and all sampling
depths. Despite the small differences between individual sites, the left panels show that all
data fall within the uncertainty bars of the reconstructed scenario of the inversion.
From 1940 to 2008 the total changes of the δ values of atmospheric $N_2O$ are (–2.2±0.2) ‰ for
$\delta^{15}N^{av}$, (–1.0±0.3) ‰ for $\delta^{18}O$, (–1.3±0.6) ‰ for $\delta^{15}N^{\alpha}$ and (–2.8±0.6) ‰ for $\delta^{15}N^{\beta}$
respectively (Fig. 2, left panels). The average linearized trends are (–0.032±0.004) ‰ $a^{-1}$ for
$\delta^{15}N$, (–0.014±0.008) ‰ $a^{-1}$ for $\delta^{18}O$, (–0.019±0.015) ‰ $a^{-1}$ for $\delta^{15}N^{\alpha}$ and (–0.041±0.020) ‰
$a^{-1}$ for $\delta^{15}N^{\beta}$. These overall trends are slightly lower compared to previous studies that used



only the data at individual sites (Ishijima et al., 2007; Bernard et al., 2006; Röckmann et al.,
2003; Sowers et al., 2002) and other studies that used data from the same period, which were
not used in the present study (Park et al., 2012). However, the differences lie well within the
combined uncertainties. We note that comparisons of average linear trends can be flawed
when the firn air records have different length and the temporal profiles do not change
linearly (see below). Trends for $\delta^{15}N^\alpha$ are smaller in magnitude than for $\delta^{15}N^\beta$, while results
from Bernard et al. (2006) showed stronger changes for $\delta^{15}N^\alpha$ than for $\delta^{15}N^\beta$. However, in that
study the trends were largely determined from measurements on young ice core samples with
comparatively higher measurement errors and scatter.
Data from two sites were not included in the multi-site inversion and are used as independent
validation of the reconstructed scenarios. The data points from Ishijima et al. (2007) (NGRIP-
01, yellow) are within the range of scenarios reconstructed by the inverse model and thus
independently validate our results. The $\delta^{15}N^{av}$ and $\delta^{18}O$ data from Sowers et al. (2002) (SP-01
in light blue and SP-95 in blue) however, agree only for the more recent atmospheric history
(Fig.2, left panels). For mean ages before 1990 most of the points are outside the uncertainty
envelopes of the multi-site reconstruction. Inter-laboratory calibration differences might be a
possible explanation for the discrepancy, but the differences are not a systematic shift, and
they are larger than offsets among laboratories that were established in the past (Sapart et al.,
2011; Kaiser et al., 2003). In fact, the data reported by Sowers et al., (2002) were actually
measured with good agreement in two different laboratories.
To evaluate our scaling approach we repeated the multi-site reconstruction using the original
non re-scaled data and the data that were re-scaled to NEEM-09 instead of DC-99 (see
Appendix C). The data rescaled to NEEM-09 gave very similar results (within uncertainties)
to the one re-scaled to DC-99 as was expected, thus the results do not depend on the choice of
the site used for re-scaling. Without re-scaling, the overall change of $N_2O$ mole fraction and
isotopic composition remained the same, but an additional decadal variability was introduced
for $\delta^{15}N^{av}$, $\delta^{15}N^\alpha$ and $\delta^{15}N^\beta$. In addition to that, the uncertainty envelopes doubled because of
the scale inconsistencies. All scaling approaches produce results that are consistent with our
preferred scaling to DC-99 within the uncertainty envelopes. We conclude that DC-99 re-
scaling removed the discrepancies that would cause larger errors if the original data were used
instead, but the re-scaling does not introduce artificial signals (see Appendix C).





The regularization of the inversion results using a rugosity factor introduces a free parameter,
which is chosen to eliminate overfitting of experimental uncertainties and which controls the
smoothness of the reconstruction. The value of this parameter is set based on a robust
generalized cross validation criterion, ensuring that the resolution obtained from the inverse
model is similar to the experimental data while taking into account the sparsity of the
measurements (Witrant and Martinerie, 2013). A sensitivity experiment where the weight of
the regularization term is increased, which leads to comparable results as linear tropospheric
histories presented in Fig. 2 was performed (Appendix C). This combined with the fact that
straight lines can be drawn within the uncertainty envelopes of the reconstructed scenarios
and the sensitivity tests (see Appendix C) indicates that the isotopic trends are not
significantly different from straight lines within the current uncertainties.

### 462    3.3    Reconstruction of the $N_2O$ emission history

Fig. 3 shows the temporal evolution of the global $N_2O$ mole fraction as inferred from the firn
air reconstruction in the top panel, and in the bottom panel the emission strength in Tg $a^{-1}$ N
calculated with the mass balance model (Section 2.5). The solid black line denotes the best
estimate scenario, which is used as input in the mass balance model. The magenta lines show
the ensemble of random scenarios generated to quantify the uncertainty of the emissions (see
Section 2.6).
The increase in the $N_2O$ mole fraction of $(32\pm1)$ nmol $mol^{-1}$ over the reconstruction period
can be explained in the mass balance model by an increase in the emissions from $(11.9\pm1.7)$
Tg $a^{-1}$ N in 1940 to $(16.4\pm1.7)$ Tg $a^{-1}$ N in 2008. The emissions increased with an increasing
trend until 1975, then the annual increase continued, but at a slower rate up to 1990, and from
then on the annual emissions have stayed approximately constant or even decreased slightly.
The corresponding changes in the mole fraction (increasing growth rate before 1990, no
increase and possibly slight decrease in the growth rate afterwards) are difficult to discern due
to the long atmospheric lifetime of $N_2O$. On average, the annual growth rate from 1995 to
2008 period is 0.7 nmol $mol^{-1}$ $a^{-1}$, corresponding to average annual emissions of 3.5 Tg $a^{-1}$ N.

### 478    3.4    The temporal evolution of the $N_2O$ isotope signatures

The results from the isotope budget calculations are presented in Fig. 4. The left panels show
the atmospheric trends. The solid black lines represent the best-fit scenarios while the dashed
black lines represent the upper and lower uncertainty envelope of the firn air reconstructions.





The magenta lines represent 50 scenarios generated randomly within the reconstructed
uncertainty range, as described in section 2.6. The middle panels show the temporal changes
in the isotope signatures of the total $N_2O$ source, with their accompanied uncertainties, as
calculated from the atmospheric mass balance model (section 2.5). The total source is split
into an assumed constant "natural" and an increasing "anthropogenic" component and the
right panels show the isotopic evolution of the "anthropogenic" component.
Results show that the average $\delta^{15}N^{av}$ of the total $N_2O$ source, over the reconstruction period, is
(−7.6±0.6) ‰ where the uncertainty is calculated using the 1σ uncertainty from the scenarios
with respect to the mean value (magenta lines). There is no statistically significant long-term
trend, but a temporal variability is observed on the decadal scale that might mask this trend.
$\delta^{15}N^{av}$ first decreased from (−6.5±0.6) ‰ in 1940 to (−8.5±0.6) ‰ in 1965, then slowly
increased again to (−6.6±0.6) ‰ in 1985, followed by another decrease to (−8.5±0.6) ‰ in
2008. These oscillations originate from the slightly curved trends in the isotopic
reconstructions for $\delta^{15}N^{av}$ in Fig. 5 (left panels).
When the source is split into a constant natural and a varying anthropogenic component, the
variability is primarily projected on the anthropogenic part and the temporal variations
increase accordingly. However, also the uncertainties increase substantially, because the
differences between the individual scenarios are attributed to only a small fraction of the total
source.
The $\delta^{15}N^{av}$ signature of the anthropogenic source has an average value of (−18.2±2.6) ‰. It
initially increases (the small initial decrease is not significant) from (−21.5±2.6) ‰ in 1940 to
(−8.6±2.6) ‰ in 1990 where it starts to slowly decrease reaching (−15.4±2.6) ‰ in 2008.
During the early part of the reconstruction period (<1970), when the "anthropogenic"
contribution was only a small fraction of the total source, the uncertainty ranges of the source
signatures are large. Therefore, the uncertainties for the early part (<1970) were excluded
from the uncertainty 1σ uncertainties from the generated scenarios. This applies to all
anthropogenic isotope signatures.
The budget calculations suggest an overall trend towards more enriched anthropogenic
emissions but the uncertainties are very large. Mathematically, this trend arises from the fact
that the isotope reconstructions yield relatively linear temporal isotope trends, whereas the
source strength increases in a strongly non-linear fashion (Fig. 4). In the beginning of the
record a small increase in the source strength needs to produce a similar absolute isotope shift



as a larger increase in the source strength in later years. This can only be satisfied by a lower
$\delta^{15}N^{av}$ value for the small "anthropogenic" emissions in the early part of the firn record. A
constant $\delta^{15}N^{av}$ source signature would result in a small temporal change in $\delta^{15}N^{av}$ of
atmospheric $N_2O$ in the beginning of the record and increasing isotope trends with increasing
emissions, similar to the exponential curves that were fit to the firn air data in Röckmann et
al. (2003).
The $\delta^{18}O$ of the total source varies within (27.2±2.6) ‰ over the entire period. $\delta^{18}O$ does not
show significant decadal scale oscillations because the reconstructed scenario for $\delta^{18}O$ is even
more strictly linear than the $\delta^{15}N^{av}$ scenario. For this reason, as explained above, in the best fit
scenario the $\delta^{18}O$ of the anthropogenic source for the initial 30 years has a more depleted
value starting with (15.2±2.6) ‰ in year 1940, reaching (31.1±2.6) ‰ in year 1975 and
remaining around this value until 2008 (Fig. 5). However, the relatively larger uncertainty
envelopes for the atmospheric history of $\delta^{18}O$ actually allow scenarios with smaller $\delta^{18}O$
changes in the beginning of the record and larger changes in the later period, which means
that the reconstruction does not exclude a constant value for the anthropogenic $\delta^{18}O$ source
signature. The available dataset thus does not allow quantifying a long-term trend in $\delta^{18}O$.
For the position dependent $^{15}N$ signatures of the total source no significant long-term trends
were detected. For $\delta^{15}N^{\alpha}$ no decadal scale variability is observed, whereas for $\delta^{15}N^{\beta}$ a
temporal variability is observed similar to the $\delta^{15}N^{av}$. The uncertainty ranges for $\delta^{15}N^{\alpha}$ and
$\delta^{15}N^{\beta}$ are about a factor 2 greater than for $\delta^{15}N^{av}$, which is due to the larger analytical error
that leads to higher uncertainties in the scenario reconstructions. $\delta^{15}N^{\alpha}$ varies in the range (–
3.0±1.9) ‰, $\delta^{15}N^{\beta}$ in the range (–11.7±2.3) ‰.
The temporal evolution of $\delta^{15}N^{\alpha}$ of the anthropogenic source looks similar to that of $\delta^{18}O$, but
with even larger variations and uncertainties with a total average of (–8.1±1.7) ‰. $\delta^{15}N^{\alpha}$
increased from (–18.2±1.7) ‰ in 1940 to an average of (-5.4±1.7) ‰ in 1975 and retained
this value until 2008. In contrast, $\delta^{15}N^{\beta}$ is similar to that of $\delta^{15}N^{av}$ with a total anthropogenic
source average of (–26.1±8.4). $\delta^{15}N^{\beta}$ initially decreases from (–19.1±8.4) ‰ to (-42.0±8.4) ‰
in 1955 only to increase again to (-10.6±8.4) ‰ in year 1990 and then decrease again to (-
26.0±8.4) ‰ in 2008.



## 4 Discussion

From the combination of the firn air reconstruction with a simple two-box model we conclude that $N_2O$ emissions increased from (11.9±1.7) Tg $a^{-1}$ N in 1940 to (16.4±1.7) Tg $a^{-1}$ N in 2008. This agrees, within uncertainties, with previous firn reconstruction studies from Ishijima et al. (2007) and Park et al. (2012) and bottom-up approaches using emission databases (Syakila and Kroeze, 2013; Kroeze et al., 1999). A more recent study by Thompson et al. (2014b) performed inversions of atmospheric measurements for 2006 to 2008 with multiple models and reported emissions of 16.1-18.7 Tg $a^{-1}$ N for 2008 in agreement with our findings.

To investigate the effect of lifetime choice on the $N_2O$ isotopic signatures (Prather et al. (2015) we performed a sensitivity study where we linearly changed the $N_2O$ lifetime from 123 years pre-industrially (≈1750) to 119 years in modern times (2008). The results are shown in Appendix D, where the effect on the emission strength and isotopic composition is discussed in detail. Results from this sensitivity study showed that the effect of a decreasing lifetime gives higher $N_2O$ emissions for year 2008 while keeping the same pre-industrial value, confirming a sensitivity in the choice of lifetime in line with Prather et al. (2015). Consequently this sensitivity only influences the anthropogenic isotopic signature by 10 %, meaning that the resulting values can increase by (2.0±1.0) ‰. The lifetime effect is most pronounced for the earliest part of the record (<1970) where the reconstruction uncertainties are larger than this systematic uncertainty.

The increase in $N_2O$ emissions over the past decades resulted in an overall decrease of all isotopic signatures of atmospheric $N_2O$ with time. The isotopic signature of the total source of $N_2O$ (Fig. 4, middle panels) is strongly depleted in all heavy isotopes compared to tropospheric $N_2O$ (Table 3), which is due to the strong enrichment associated with the removal in the stratosphere. In Table 3 the isotopic composition for the pre-industrial period (≈1750) ($\delta_{nat,pi}$) is compared with the derived anthropogenic source signature derived from our multi-site reconstruction ($\delta_{anth}$, averaged from 1940 to 2008). The results show that the anthropogenic source is more depleted in heavy isotopes than the natural one for all signatures, confirming results from previous studies that used forward firn air modelling on measurements from individual sites (Park et al., 2012; Ishijima et al., 2007; Röckmann et al., 2003).



Anthropogenic $N_2O$ emissions are dominated by agricultural soil (70 %) with smaller
contributions from automobiles, coal combustion, biomass burning and industry. Oceanic
emissions were previously assumed to be only natural. However, the latest IPCC Assessment
Report (IPCC, ch.6, 2013) for the first time breaks down oceanic emissions into a natural and
an anthropogenic component, e.g. due to atmospheric N deposition to rivers (Syakila and
Kroeze, 2011; Duce et al., 2008; Kroeze et al., 2005), thus estimating the anthropogenic
component of oceanic $N_2O$ emissions to amount to 1 Tg a$^{-1}$ N.
$N_2O$ emitted from agricultural soils and biomass burning is more depleted in $\delta^{15}N$ and $\delta^{18}O$
than the tropospheric background (Park et al., 2011; Goldberg et al., 2010; Ostrom et al.,
2010; Tilsner et al., 2003; Perez et al., 2001; 2000) while $N_2O$ emitted from other minor
sources, such as automobiles, coal combustion and industry, has values closer to tropospheric
$N_2O$ values (Syakila and Kroeze, 2011; Toyoda et al., 2008; Ogawa and Yoshida, 2005a;
2005b).
Qualitatively, an increase of strongly depleted agricultural emissions in the first part of our
reconstruction, followed by a decreasing relative contribution from agriculture and increasing
contributions from more enriched sources like industry, automobiles and coal combustion,
can explain the reconstructed changes of isotope signatures of both the total source and the
anthropogenic component. The global $N_2O$ budget study from Syakila and Kroeze (2011)
indicates that agricultural emissions were 78 % of the total during the 1940-1980 period with
little input from industry, vehicle exhaust and coal combustion. After 1980 the relative share
of agricultural emissions dropped to 64 %, while the other sources increased, supporting our
suggestion.
Additional evidence for potential changes in the $N_2O$ source composition between the pre-
industrial and present atmosphere may be derived from the position-dependent $^{15}N$ signatures,
quantified by the $^{15}N$ site preference. Table 3 shows that the difference in the $\delta^{15}N^{av}$ signature
between the pre-industrial and the anthropogenic source derived from our reconstruction is
primarily due to a change at position $\delta^{15}N^{\beta}$, whereas $\delta^{15}N^{\alpha}$ remains relatively constant. This is
reflected by a larger difference in $\delta^{15}N^{sp}$ between natural and anthropogenic emissions, which
could indicate a temporal change in production processes.
Sutka et al. (2006) suggested that there may be two distinct classes of $N_2O$ sources with
different $\delta^{15}N^{sp}$. $N_2O$ produced during nitrification and fungal denitrification had a high $\delta^{15}N^{sp}$
of (33±5) ‰ and $N_2O$ from denitrification and nitrifier denitrification had a low $\delta^{15}N^{sp}$ of





(0±5) ‰. Park et al., (2012) used these two endmembers to calculate a change in the relative
fractions of these source classes over time based on their firn air data. Although this approach
is strongly simplified and several other sources and factors may contribute (Toyoda et al.,
2015), we use the results from our box model calculations (Table 3) in a similar way to
estimate the fraction of the two source categories according to the following simple mass
balance calculation:
$$F_{\text{high}} = \frac{\delta^{15}\text{N}_{\text{meas}}^{\text{sp}} - \delta^{15}\text{N}_{\text{low}}^{\text{sp}}}{\delta^{15}\text{N}_{\text{high}}^{\text{sp}} - \delta^{15}\text{N}_{\text{low}}^{\text{sp}}} \qquad\qquad (11)$$

This returns a fractional contribution of the $\delta^{15}\text{N}_{\text{high}}^{\text{sp}}$ component of (19±4) % to the total pre-
industrial emissions and (35±11) % to the total present source. The errors were derived by
propagating the errors of the $\delta^{15}\text{N}^{\text{sp}}$. endmembers and $\delta^{15}\text{N}_{\text{meas}}^{\text{sp}}$ as given above. We note that
the errors associated with the precise isotopic composition of the endmembers are correlated
if $\delta^{15}\text{N}^{\text{sp}}$ for the two endmembers remain relatively constant. Therefore, the change in the
relative fraction of the two categories is likely better constrained than the absolute values.
Splitting the total present emission strength into a natural (pre-industrial, 11.0 Tg a$^{-1}$ N) and
anthropogenic (5.4 Tg a$^{-1}$ N) component, we derive a fraction of the $\delta^{15}\text{N}_{\text{high}}^{\text{sp}}$ component
(which includes nitrification) of (54±26) % for the "anthropogenic" emissions. This is another
piece of evidence for agricultural sources being the main contributor to the N$_2$O increase,
because nitrification-dominated agricultural emissions can be associated with the $\delta^{15}\text{N}_{\text{high}}^{\text{sp}}$
component.
The temporal changes of the derived fraction of nitrification are in good qualitative agreement
with the results from Park et al. (2012), who reported a change of (13±5) % from 1750 to
(23±13) % today. However, the absolute numbers derived from our study are higher than the
results from Park et al. (2012). The difference is due to the fact that different apparent isotope
fractionations during stratospheric removal ($\varepsilon_{\text{app}}$) are used in the mass balance model (Table
3; eq. 7,8). In this study we used the averaged lowermost stratospheric apparent isotope
fractionations from Kaiser et al. (2006), which we consider more representative than the
numbers used by Park et al. (2012). Using different values for $\varepsilon_{\text{app}}$ causes a shift in the
isotopic source signatures from the mass balance model. The choice of this value thus adds a
systematic source of uncertainty to the absolute value of the $\delta^{15}\text{N}_{\text{high}}^{\text{sp}}$ fractions reported above
($F_{\text{high}}$).



Nevertheless, this systematic uncertainty should not alter the overall *change* in $F_{high}$ from pre-
industrial to modern times and the results from our multi-site reconstruction of the isotopic
composition of $N_2O$ thus confirm the suggestion by Park et al. (2012) that the relative
importance of the high-SP component (presumably nitrification) has increased with increasing
mole fraction since pre-industrial times.

## 641    5    Conclusions

The temporal evolution of the total $N_2O$ emission fluxes and the source isotopic composition
have been estimated in a top-down approach using a multi-site reconstruction of $N_2O$ mole
fraction and isotopic composition from 6 firn air samplings at 5 different Arctic and Antarctic
locations in a two-box model. The results from a mass balance model constraints the source
strength and suggest a total increase in $N_2O$ emissions of ($4.5\pm1.7$) Tg $a^{-1}$ N between the 1940
and 2008 due to anthropogenic processes. This agrees with previous top-down estimates, but
deviates from bottom-up model estimates, which suggest higher $N_2O$ emission increases. A
significant source of the uncertainty in such top-down estimates is a possible change in the
$N_2O$ lifetime over the reconstruction period, which we have quantified following the recent
results from Prather et al. (2015).
The reconstruction of mole fraction and isotopic composition was used to investigate
temporal changes in the isotopic signature of $N_2O$ emissions over the study period. The
average total source for $\delta^{15}N^{av}$ and $\delta^{15}N^{\beta}$ shows no statistically significant long-term trend but
likely significant decadal scale variability. For $\delta^{18}O$ and $\delta^{15}N^{\alpha}$ of the total $N_2O$ source, no
significant temporal changes can be detected with the present dataset because the
uncertainties are large, especially in the beginning of the reconstruction period.
When the total source is split into a constant natural and a varying anthropogenic component,
the reconstruction of the $\delta$ values of the anthropogenic source indicates a significant increase
of $\delta^{15}N^{av}$ from the early to the modern part of the record. This originates from the near-linear
isotope histories of the best guess scenario, which would imply that small emissions in the
early part had a similar absolute effect on the $\delta$ values as stronger emissions in the latter part.
A similar effect for $\delta^{18}O$ is likely, but not significant given the larger uncertainties for this
signature.
Nevertheless, the isotope signal in $\delta^{15}N^{av}$ may also be a signal for changing source
contributions over time. Bottom-up models suggest that $N_2O$ emitted from agricultural soils



was the dominant contributor to the anthropogenic $N_2O$ increase in the first decades. Smaller
contributions due to emissions from more enriched sources, like industry, automobiles and
coal combustion increased, which may have contributed to an isotope enrichment of the
emissions, which is not detectable within the error bars for the other isotope signatures.
Results from the mass balance model yield an increase in $^{15}N$ site preference between the pre-
industrial and modern total $N_2O$ source. The increase in $\delta^{15}N^{sp}$ of $(16\pm11)$ % between the pre-
industrial and modern source is in qualitative agreement with increased emissions from
nitrification processes associated with agriculture.
**Acknowledgements**
We thank the teams involved in the firn air sampling at NEEM site during the 2008 and 2009
field seasons. NEEM is directed and organized by the Centre of Ice and Climate at the Niels
Bohr Institute, University of Copenhagen, Denmark and the US National Science Foundation,
Office of Polar Programs. It is supported by funding agencies and institutions in Belgium
(FNRS-CFB and FWO), Canada (NRCan/GSC), China (CAS), Denmark (FIST), France
(IPEV, CNRS/INSU, CEA and ANR), Germany (AWI), Iceland (RannIs), Japan (NIPR),
Korera (KOPRI), The Netherlands (NWO/ALW and NWO/NPP), the United Kingdom
(NERC NE/F021194/1) and the USA (US NSF, Office of Polar Programs). This project was
financially supported by the Dutch Science Foundation (NWO), projects 851.30.020 &

685 865.07.001.



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



Table 1. Site information on the drilling locations of the North Greenland Ice core Project
(NGRIP-01), Berkner Island (BKN-03), North Greenland Eemian Ice drilling Project
(NEEM-EU-08, NEEM-09), Dome Concordia (DC-99) and Dronning Maud Land (DML-98),
where firn air samples were collected, and two key meteorological variables of each site.

| Site | Location | Mean annual temperature (°C) | Surface accumulation rate (water equivalent) (cm a$^{-1}$) | Sampling year |
|---|---|---|---|---|
| NGRIP-01 | 75° N 42° W | -31 | 20 | 2001 |
| BKN-03 | 79° S 45° W | -26 | 13 | 2003 |
| NEEM-EU-08 | 77.4° N 51.1° W | -29 | 22 | 2008 |
| NEEM-09 | 77.4° N 51.1° W | -29 | 22 | 2009 |
| DC-99 | 75° S 123° E | -53 | 3 | 1999 |
| DML-98 | 75° S 65° E | -38 | 6 | 1998 |


Table 2. Detailed information in the mole fraction and the isotopic composition of the
laboratory reference gases used for correcting each set of firn air samples.

| Site | Sampling year | Mole fraction (nmol mol$^{-1}$) | $\delta^{15}N^{av}$ (‰) | $\delta^{18}O$ (‰) | $\delta^{15}N^{\beta}$ (‰) | $\delta^{15}N^{\alpha}$ (‰) |
|---|---|---|---|---|---|---|
| NGRIP-01 | 2001 | 318 | 6.64 | 44.61 | -2.79 | 16.07 |
| BKN-03 | 2003 | 318 | 6.64 | 44.61 | -2.79 | 16.07 |
| NEEM-EU-08 | 2008 | 324 | 6.22 | 44.40 | -3.08 | 15.52 |
| NEEM-09 | 2009 | 318 | 6.38 | 44.92 | -2.66 | 15.41 |
| DC-99 | 1999 | 318 | 6.64 | 44.61 | -2.79 | 16.07 |
| DML-98 | 1998 | 318 | 6.64 | 44.61 | -2.79 | 16.07 |



Table 3. Stratospheric isotope fractionation ($\varepsilon_{app}$) used in the mass balance model, and
isotopic composition of the natural and anthropogenic source, and the respective results from
Park et al. (2012).

| | $\varepsilon_{app}$( ‰) this study* | $\varepsilon_{app}$( ‰) Park et al., 2012* | $\delta_{atm,pi}$ ( ‰) Park et al., 2012 | $\delta_{nat,pi}$ ( ‰) this study* | $\delta_{nat,pi}$ ( ‰) Park et al., 2012 | $\delta_{anth}$ ( ‰) this study* | $\delta_{anth}$( ‰) Park et al., 2012** |
|---|---|---|---|---|---|---|---|
| $\delta^{15}N$ | -16.2 | -14.9 | 9.3±0.2 | -5.2±0.2 | -5.3±0.2 | -18.2±2.6 | -15.6±1.2 |
| $\delta^{18}O$ | -13.4 | -13.3 | 45.5±0.2 | 33.1±0.2 | 32.0±0.2 | 27.2±2.6 | 32.0±1.3 |
| $\delta^{15}N^{\alpha}$ | -23.0 | -22.4 | 18.8±1.0 | -1.9±1.0 | -3.3±1.0 | -8.1±1.7 | -7.6±6.2 |
| $\delta^{15}N^{\beta}$ | -9.4 | -7.1 | -0.6±1.1 | -8.3±1.1 | -7.5±1.1 | -26.1±8.4 | -20.5±7.1 |
| $\delta^{15}N^{sp}$ | - | - | 19.4±1.5 | 6.4±1.5 | 4.2±1.5 | 18.0±8.6 | 13.1±9.4 |

*$\varepsilon_{app}$ values used in this study are averaged values from the lower stratosphere from Kaiser et al. (2006) and $\varepsilon_{app}$
values from Park et al. (2012) were used from Park et al. (2004). $\delta_{atm,pi}$ values are from Park et al. (2012) who
also calculated $\delta_{nat,pi}$ and $\delta_{anth}$ in a two-box model. Here, the $\delta_{anth}$ values are the averaged values over the whole
investigated period.










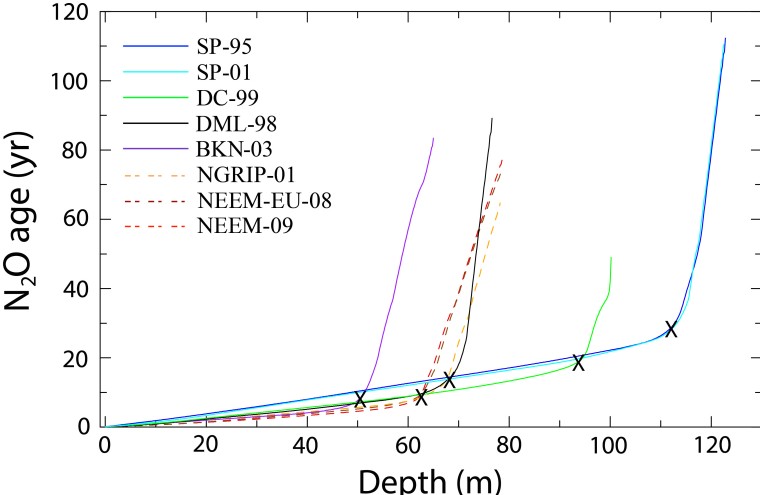


Figure 1. N₂O mean ages in firn versus depth. The dashed lines represent the sites from the
NH (North Greenland Ice-core Project [NGRIP-01], North Eemian Ice-core Project  [NEEM-
09, NEEM-EU-08]) and the solid lines the SH sites (South Pole [SP-01, SP-95], Dome C
[DC-99], Dolomite [DML-98] and Berkner Island [BKN-03]). The numbers accompanying
the sites are the corresponding drilling years. Marker X indicates the transition between the
firn diffusive zone and the bubble close-off zone for each site. Dashed orange line NGRIP-01,
dashed brown NEEM-EU-08, dashed red NEEM-09, purple line BKN-03, black DML-98,
green DC-99, blue SP-95 and light blue SP-01.



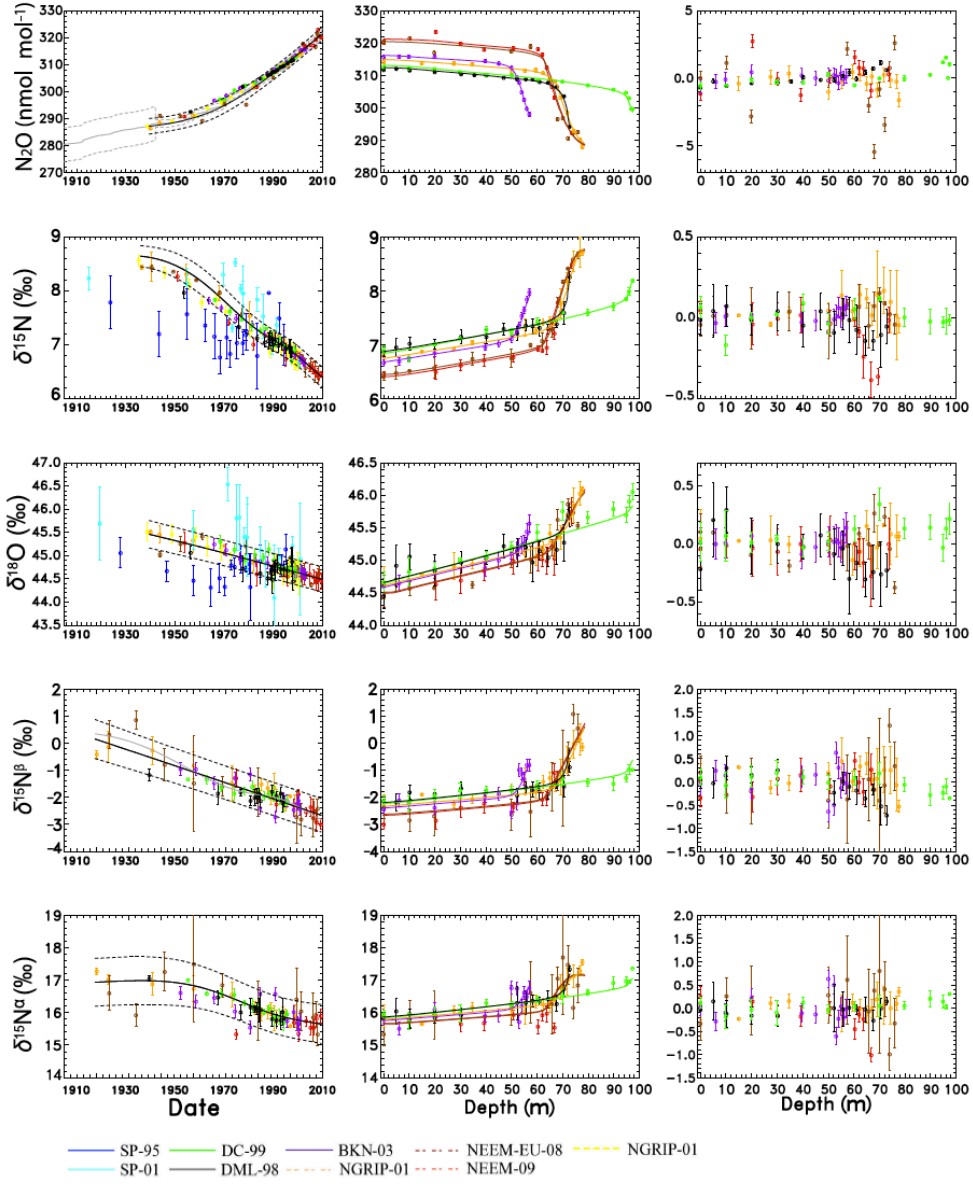









Figure 2. Left: Reconstructed atmospheric scenarios (black solid line with dashed lines
indicating the 2 σ uncertainty intervals) and results of the firn air samples (corrected for firn
fractionation) plotted at their respective assigned mean age. Middle: corresponding depth
profiles, symbols show the measurements and solid lines the results of the forward model
using the best estimate scenario as input. Right: model data discrepancies. Orange: NGRIP-01
(Bernard et al, 2006), Yellow: NGRIP-01 (Ishijima et al., 2007), Brown: NEEM-EU-08, Red
NEEM-09, Purple: BKN-03, Black: DML-98, Green: DC-99, Blue: SP-95 and Light Blue:
SP-01. Data from NGRIP-01 (Ishijima et al., 2007), SP-95 and SP-01 were not used in the
atmospheric reconstruction and are only plotted for comparison purposes here.

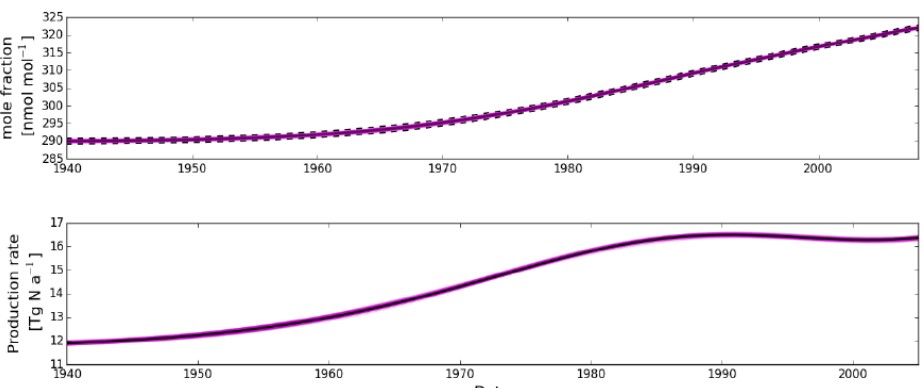


Figure 3: Top panel. $N_2O$ mole fraction history from the multisite reconstruction (solid black
line with uncertainty envelopes as dashed black lines) and the scenarios within the uncertainty
envelopes that were used in the mass balance model (magenta lines) to evaluate the
uncertainties of the atmospheric modelling results.
Bottom panel. $N_2O$ production rate as calculated from the mass balance model. The solid
black line represents the result for the best fit reconstruction while magenta lines represent the
results for the individual scenarios from the top panel.






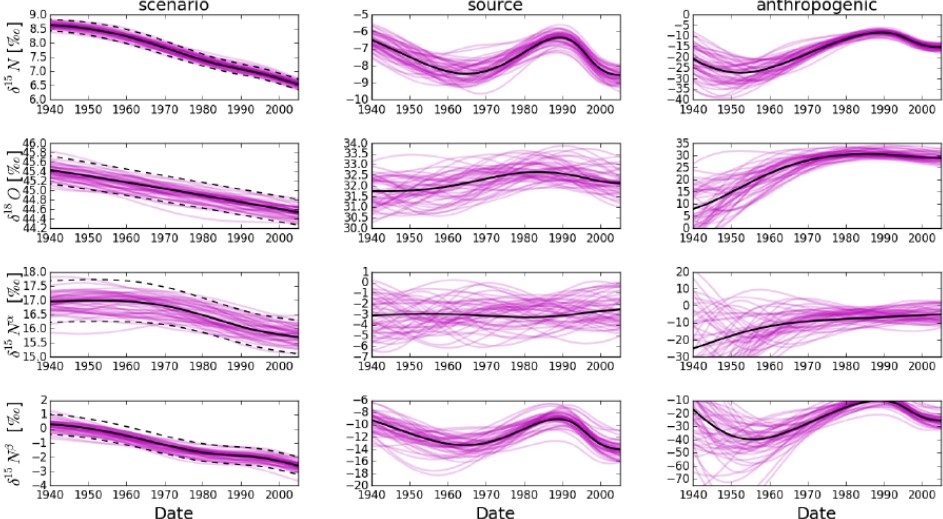


Figure 4: Left panels: Historic evolution of $\delta^{15}N$, $\delta^{18}O$, $\delta^{15}N^{\alpha}$ and $\delta^{15}N^{\beta}$ in $N_2O$ as derived
from the firn air reconstruction. Middle panels: isotope signatures of the total emitted $N_2O$.
Middle panels: isotope signatures of the anthropogenic source, respectively. The solid black
line represents the best-fit scenario while the dashed ones represent the respective
uncertainties as determined by the reconstruction method. Magenta lines represent the
emissions that are required to produce the magenta $N_2O$ histories in the left panels.













**Appendix A: Effect of firn fractionation on isotopic composition**





Figure A1: Effect of firn fractionation on $N_2O$ isotopic composition in firn. Original
measurements are plotted as stars, data corrected for firn fractionation are plotted as circles
with error bars. The right hand side shows Southern hemisphere sites, orange: NGRIP-01
(Bernard et al, 2006), yellow: NGRIP-01 (Ishijima et al, 2007), brown: NEEM-EU-08, red
NEEM-09 and the left hand side shows Northern hemisphere sites, purple: BKN-03, black
DML-98, green DC-99, blue SP-95 and light blue SP-01

























**Appendix B: Data processing**
In this study isotope deltas ($\delta$) are used to denote the relative $^{15}N/^{14}N$ and $^{18}O/^{16}O$ ratio
difference of $N_2O$ in firn air with respect to a standard reference,
$\delta^{15}N = \dfrac{R_{sample}}{R_{standard}} - 1$                 (1)
where R represents the $^{15}N/^{14}N$ or $^{18}O/^{16}O$ abundance ratio of a standard or a sample. $\delta^{15}N$
values are reported relative to $^{15}R$ of atmospheric $N_2$, $\delta^{18}O$ values relative to $^{18}R$ of Vienna
Mean Standard Ocean Water (VSMOW). The $^{15}N/^{14}N$, $^{18}O/^{16}O$ and position dependent
$^{15}N/^{14}N$ isotope ratios were derived from measurement of the $m/z$ 45 / $m/z$ 44, $m/z$ 46 / $m/z$ 44
and $m/z$ 31 / $m/z$ 30 ion current ratios according to Kaiser et al., (2008), assuming a constant
$^{17}O$ excess of 0.9 ‰.
There is a disagreement between reported trends of the position dependent $\delta^{15}N^{av}$ values
reported in the literature from firn air on the one hand and archived air samples on the other
hand (Park et al., 2012; Ishijima et al., 2007; Bernard et al., 2006; Röckmann and Levin,
2005; Röckmann et al., 2003; Sowers et al., 2002). In principle the temporal trend measured
directly on archived air samples should be fully consistent with top firn air samples of the
various data sets, which were collected over a decade or more, since the air in the diffusive
zone is not very old. However, this is not the case. Using the high-precision determination of
the temporal trend of the $N_2O$ isotope signatures on archived air samples from Röckmann and
Levin (2005) as reported in section 2.4 we rescale the different firn profiles to match this
trend in the diffusive zone by interpolating the measurements from the diffusive zone of all
sites to DC-99 ($\delta_{INT}$). By using the firn model – assigned mean age of each sample, The
maximum age difference from diffusive zone to surface corresponds to $\Delta age = \Delta_{DC\ t-t_0} = 10$ a.
Below you can find the equations used:
$\delta_{INT} = \delta_{t-t_0} - \delta_{DC\ t-t_0} + m\ (\Delta_{t-t_0} - \Delta_{DC\ t-t_0})$         (2)
$\delta_{Final} = \delta_{meas} - (\delta_{exp} - \delta_{INT})$              (3)
Where $m$ is the slope connecting the two points we want to interpolate. The applied scaling
($\delta_{Final}$) is given in the Table B1 below. To bring the data to the most recent international scale,
NOAA-2006A, we used an equation extracted from a correlation between a scale ratio of
NOAA-2006A to CSIRO versus the mole fraction of $N_2O$. The correlation showed higher



scale ratio for low fraction values and lower scale ratio for higher mole fraction values.  The
equation extracted is given below:
$y(\text{NOAA-2006}) = -1.535 \times 10^{-4}\, y^2(\text{CSIRO}) + 1.045\, y(\text{CSIRO})$          *(4)*



Table B1. Implemented scaling for $N_2O$ mole fraction and isotopic composition. The re-
scaled average was extracted from the diffusivity zone for each site, which corresponds to the
top 50 m. The expected trends are averaged values from CSIRO
(http://www.csiro.au/greenhouse-gases) for the last 30 years for the mole fraction and
measured trends from Röckmann and Levin (2005) for the isotopic composition. The rather
large corrections to the isotope data from the SP-01 and SP-95 drillings are likely due to inter-
laboratory scale differences.

| Site | $y(N_2O)(nmol\ mol^{-1})$ | | |
| | Re-scaled average | Expected trend change | Correction |
| --- | --- | --- | --- |
| DML-98 | 0.09±0.29 | -0.80±0.06 | -0.89±0.32 |
| NGRIP-01(Bernard) | 3.39±0.54 | 1.60±0.06 | -1.79±0.54 |
| NGRIP-01 (Ishijima) | 4.12±0.32 | 1.60±0.06 | -2.52±0.32 |
| BKN-03 | 3.47±0.22 | 3.20±0.06 | -0.27±0.23 |
| NEEM-EU-08 | 3.57±1.81 | 7.20±0.06 | 3.63±1.81 |
| NEEM-09 | 8.84±1.82 | 8.00±0.06 | -0.84±1.82 |

| Site | $\delta^{15}N\ (\permil)$ | | |
| | Re-scale average | Expected trend change | Correction |
| --- | --- | --- | --- |
| SP-95 | 1.43±0.56 | 0.16±0.00 | -1.27±0.56 |
| DML-98 | -0.18±0.12 | 0.04±0.00 | 0.22±0.12 |
| SP-01 | 0.22±0.22 | -0.08±0.00 | -0.30±0.22 |
| NGRIP -01(Bernard) | -0.18±0.07 | -0.08±0.00 | 0.10±0.07 |
| NGRIP -01 (Ishijima) | 0.17±0.13 | -0.08±0.00 | -0.25±0.13 |
| BKN-03 | -0.17±0.12 | -0.16±0.00 | 0.01±0.12 |
| NEEM-EU-08 | -0.63±0.15 | -0.36±0.00 | 0.27±0.15 |
| NEEM-09 | -0.43±0.05 | -0.40±0.00 | -0.03±0.05 |



| Site | $\delta^{18}O$ (‰) | | |
|---|---|---|---|
| | Re-scale average | Expected trend change | Correction |
| SP-95 | -0.88±0.27 | 0.08±0.00 | 0.96±0.27 |
| DML-98 | 0.26±0.15 | 0.02±0.00 | -0.24±0.15 |
| SP -01 | 0.74±0.62 | -0.04±0.00 | -0.78±0.62 |
| NGRIP-01 (Bernard) | -0.08±0.05 | -0.04±0.00 | 0.04±0.05 |
| NGRIP-01 (Ishijima) | -0.17±0.12 | -0.04±0.00 | 0.13±0.12 |
| BKN-03 | 0.02±0.06 | -0.08±0.00 | -0.10±0.06 |
| NEEM-EU-08 | -0.21±0.15 | -0.19±0.00 | 0.02±0.15 |
| NEEM-09 | 0.28±0.04 | -0.21±0.00 | -0.49±0.04 |

| Site | $\delta^{15}N^{\beta}$ (‰) | | |
|---|---|---|---|
| | Re-scale average | Expected trend change | Correction |
| DML-98 | -0.41±0.20 | 0.06±0.02 | 0.47±0.20 |
| NGRIP-01 (Bernard) | -0.10±0.25 | -0.13±0.02 | -0.02±0.25 |
| BKN-03 | -0.53±0.30 | -0.26±0.02 | 0.27±0.30 |
| NEEM-EU-08 | -0.33±0.27 | -0.58±0.02 | -0.25±0.27 |
| NEEM-09 | -0.14±0.17 | -0.64±0.02 | -0.50±0.17 |

| Site | $\delta^{15}N^{\alpha}$ (‰) | | |
|---|---|---|---|
| | Re-scale average | Expected trend change | Correction |
| DML-98 | 0.09±0.11 | 0.01±0.02 | -0.08±0.11 |
| NGRIP-01 (Bernard) | -0.26±0.19 | -0.03±0.02 | 0.23±0.19 |
| BKN-03 | 0.19±0.32 | -0.06±0.02 | -0.25±0.32 |
| NEEM-EU-08 | -0.61±0.35 | -0.13±0.02 | 0.48±0.35 |
| NEEM-09 | -0.72±0.16 | -0.14±0.02 | 0.58±0.16 |





**Appendix C: Atmospheric reconstruction re-scaled to NEEM-09 and without data re-scaling**

Figure C1. Results of the firn data evaluation (similar to Figure 2) using the data without re-scaling as indicated in the text, Orange: NGRIP-01 (Bernard et al, 2006), Yellow: NGRIP-01





(Ishijima et al, 2007), Brown: NEEM-EU-08, Red: NEEM-09, Purple: BKN-03, Black: DML-
98, Green: DC-99, Blue: SP-95 and Light Blue: SP-01.

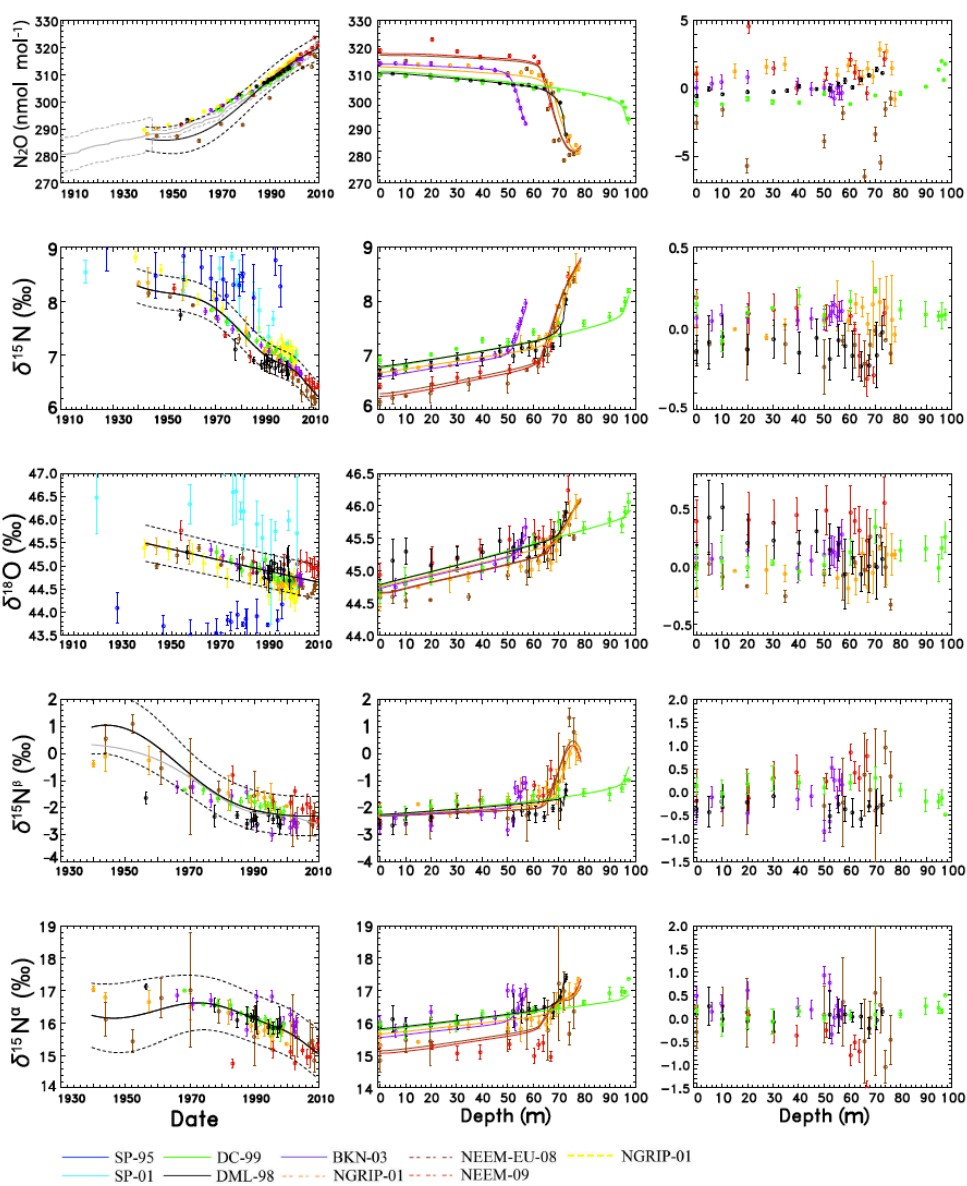


Figure C2. Results of the firn data evaluation (similar to Figure 2)  using the data re-scaled to
the NEEM-09 site. Colours as in Fig. C1.

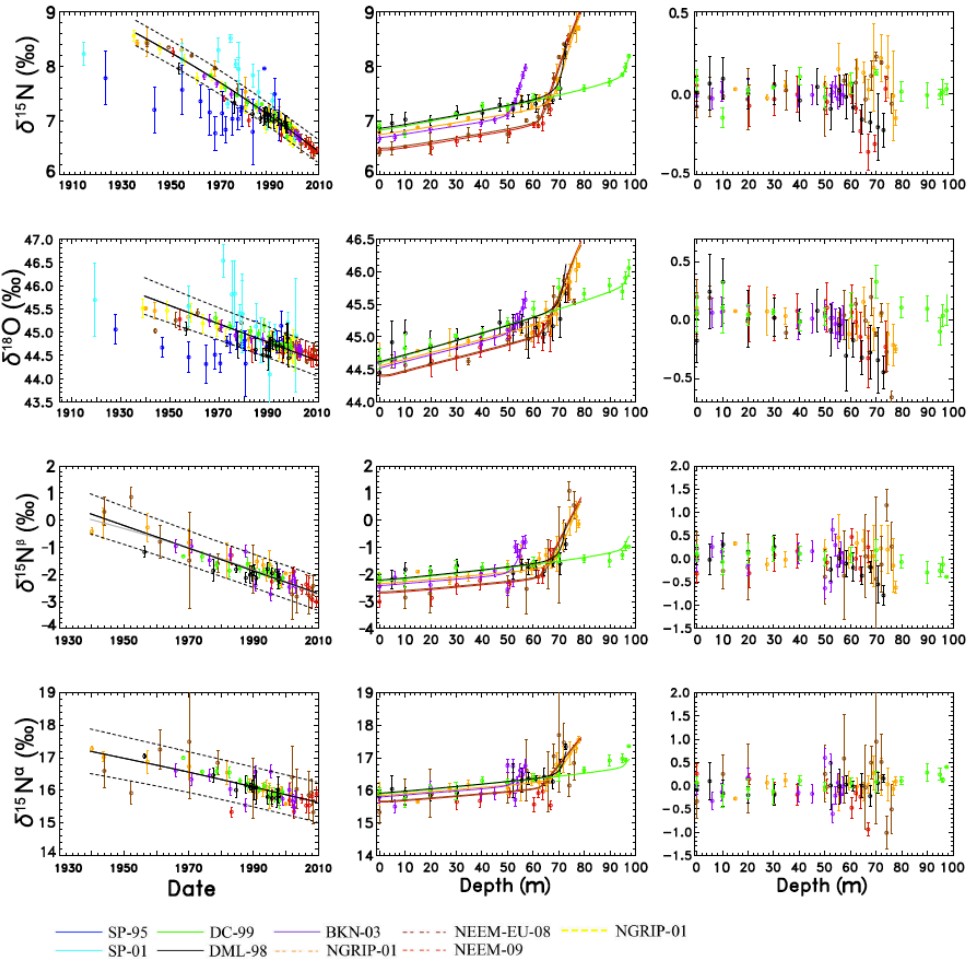


Figure C3. Sensitivity test for the regularization term. Reconstructed atmospheric scenarios (left), corresponding fit of the firn data (centre) and model data discrepancies (right). The best reconstructed scenarios are shown as the black continuous lines, with model derived uncertainties ($2\sigma$) in dashed lines. Colours as in Fig. C1.






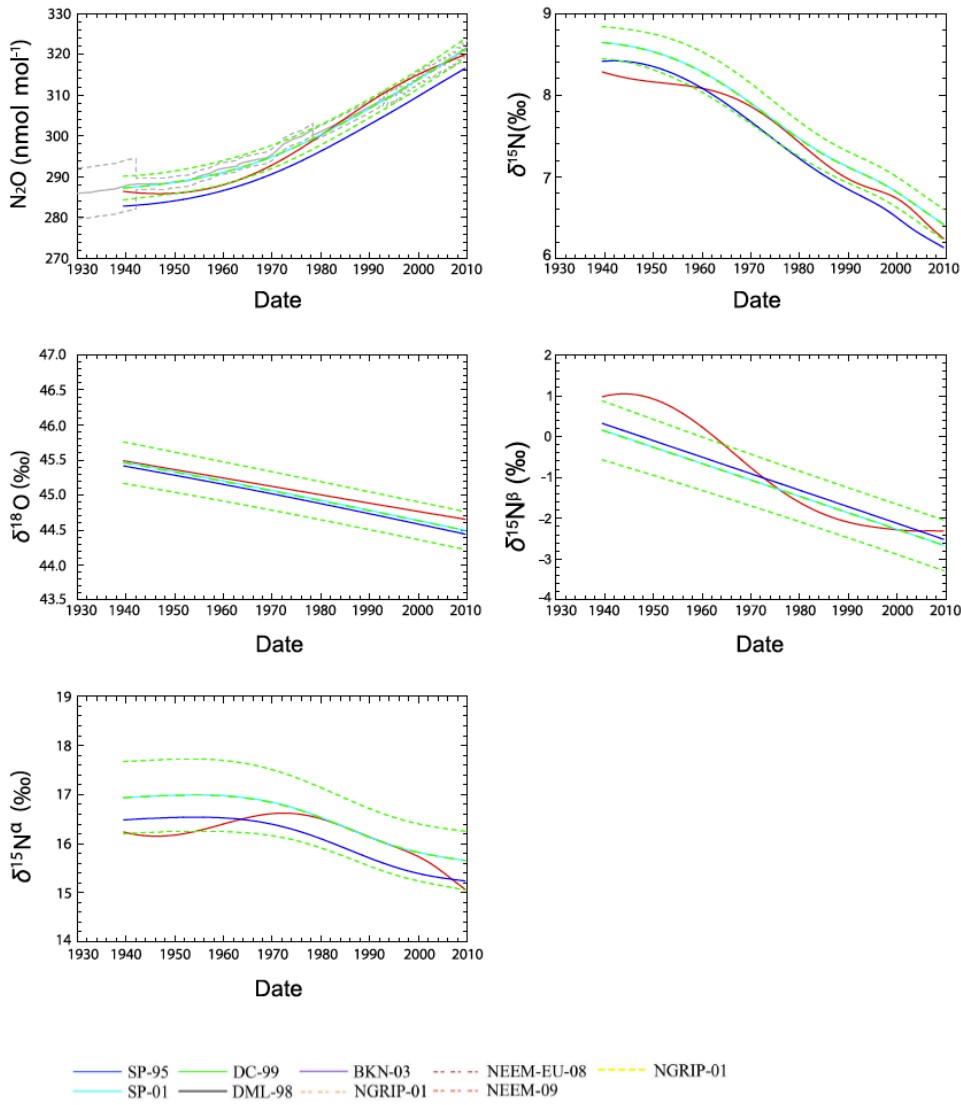


Figure C4. Comparison of the atmospheric reconstructions between different re-scaling methods. Solid and dashed green lines are the scenarios from data re-scaled to DC-99 used in this study. Solid red lines are the best-case scenario for the non re-scaled data and solid blue lines are the best-case scenarios from the data re-scaled to NEEM-09. The latter data series is shifted because of a calibration offset. When this is corrected for the data superimposes the green lines as expected.






**Appendix D: Sensitivity of the reconstructed $N_2O$ emissions and isotopic**
**signatures on $N_2O$ lifetime.**
For the default calculations with the mass balance model a constant lifetime for $N_2O$ was
used. A recent study from Prather et al. (2015), though, highlighted that top-down model
calculations are sensitive to changes in the $N_2O$ lifetime. To quantify the effect on our results
we performed a sensitivity test where we linearly changed the $N_2O$ lifetime from pre-
industrial to modern times from 123 a in 1700 to 119 a in 2008. The results are shown in
Figures D1 and D2 below.
In Figure D1 the $N_2O$ atmospheric budget is re-calculated and compared with the results when
the constant lifetime of $123^{+29}_{-19}$ a is used. In year 1940 the $N_2O$ emissions are (12.3±2.7) Tg $a^{-1}$
N and (17.0±1.7) Tg $a^{-1}$ N in year 2008 with a total increase of (4.7±1.7) Tg $a^{-1}$ N. When
keeping the lifetime constant, the results for the same years are (11.9±1.7) Tg $a^{-1}$ N and
(16.4±1.7) Tg $a^{-1}$ N with a total increase of (4.5±1.7) Tg $a^{-1}$ N. This shows that there is a
sensitivity on the choice of lifetime for our mass balance model on the $N_2O$ atmospheric
budget as was indicated by Prather et al. (2015).
The $N_2O$ source isotopic signature shows no significant change with the choice of lifetime
giving similar average source values for all source signatures as for when using a constant
lifetime of $123^{+29}_{-19}$ a.
On the other hand, the $N_2O$ average anthropogenic source signature displays a sensitivity in
the choice of lifetime returning values (−15.9±2.6) ‰, (28.5±2.6) ‰, (−7.2±1.7) ‰ and (−
22.8±8.4) ‰ for $\delta^{15}N^{av}$, $\delta^{18}O$, $\delta^{15}N^{\alpha}$ and $\delta^{15}N^{\beta}$ respectively. This agrees within combined
errors with the total average values of (−18.2±2.6) ‰, (27.2±2.6) ‰, (−8.1±1.7) ‰ and (−
26.1±8.4) ‰ for $\delta^{15}N^{av}$, $\delta^{18}O$, $\delta^{15}N^{\alpha}$ and $\delta^{15}N^{\beta}$ respectively when a constant $123^{+29}_{-19}$ a lifetime
is used. On average, the $N_2O$ anthropogenic signature results can differ by 10 % when a
different lifetime is chosen, which is equivalent to a (2.0±1.0) ‰ difference in the final
anthropogenic values.





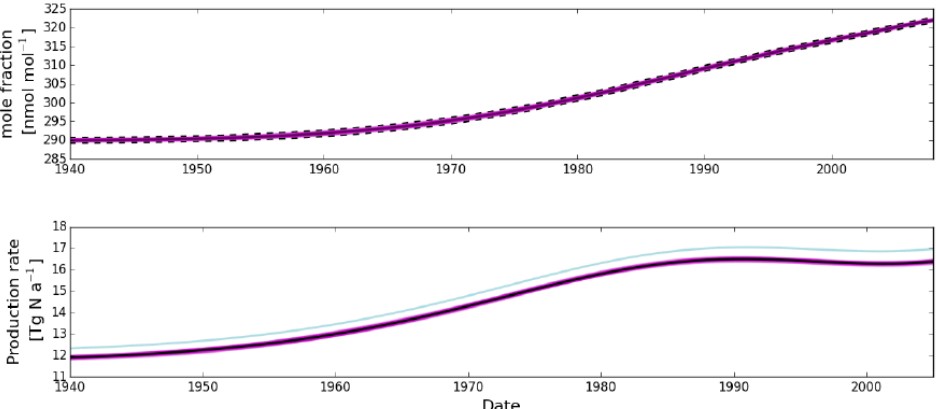


Figure D1: Top panel. $N_2O$ mole fraction history from the multisite reconstruction (solid
black line with uncertainty envelopes as dashed black lines) and the scenarios within the
uncertainty envelopes that were used in the mass balance model (magenta lines) to evaluate
the uncertainties of the atmospheric modelling results.
Bottom panel. $N_2O$ production rate as calculated from the mass balance model assuming a
change in the lifetime from 123 a in 1700 to 119 a in 2008 (relative change similar to Prather
et al., 2015) in light blue. The solid black line represents the result for the best fit
reconstruction while magenta lines represent the results for the individual scenarios from the
top panel (lifetime kept constant at $123^{+29}_{-19}$ a) as used in the main paper.











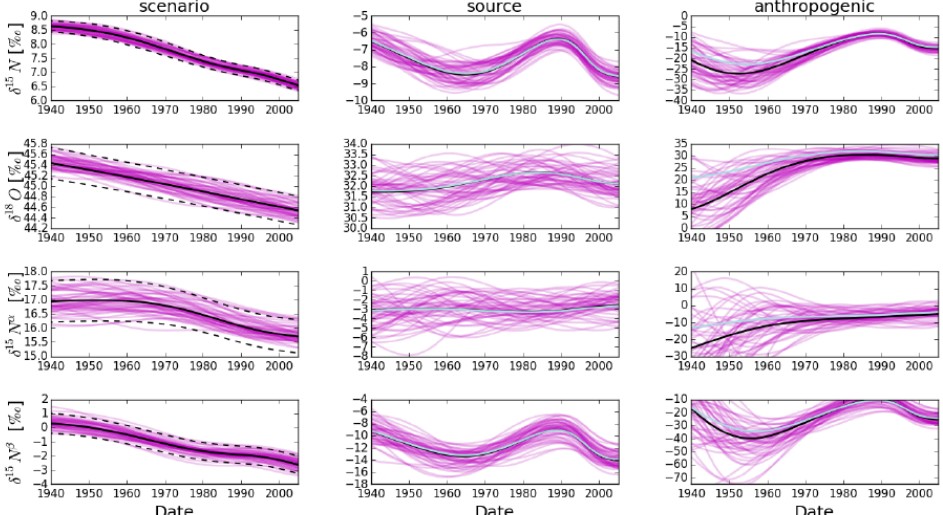


Figure D2: Left panels: Historic evolution of $\delta^{15}N$, $\delta^{18}O$, $\delta^{15}N^{\alpha}$ and $\delta^{15}N^{\beta}$ in $N_2O$ as derived
from the firn air reconstruction. The solid black line represents the best-fit scenario while the
dashed ones represent the respective uncertainties as determined by the reconstruction
method. Magenta lines represent the emissions that are required to produce the magenta $N_2O$
histories in the left panels. Middle and right panels: Isotope signatures of the total emitted
$N_2O$ and anthropogenic source respectively assuming a change in the lifetime from 123 a in
1700 to 119 a in 2008 (relative change similar to Prather et al., 2015) in light blue. The solid
black line represents the result for the best fit reconstruction while magenta lines represent the
results for the individual scenarios from the top panel (lifetime kept constant at $123^{+29}_{-19}$ a) as
used in the main paper.