# Peer review of "Constraining N2O emissions since 1940 using firn air isotope measurements in"

_Atmospheric Chemistry and Physics, 2016_

## Referee Comment (RC1) · T. Rahn (Referee) · 1 Nov 2016

The authors provide a synthesis of firn air N2O isotopic data sets in an attempt to better constrain our understanding of the natural and anthropogenic components of N2O emissions. The synthesis of these geographically diverse records is acceptable since the long atmospheric lifetime of N2O allows the assumption that all sites experience an N2O history that is identical within the precision of the measurement capabilities and indeed the authors have done a fine job in the firn air modeling and in the scaling of the different data sets. I do however have issues with some of the follow on modeling and discussion which I itemize in the line by line comments below. I recommend publication following revision. Line 25. The 1940 atmospheric mole fraction, and hence

the growth rate, differs from that determined by Battle et al., 1996 and to a lesser degree with Machida et al., 1995, please discuss (in the discussion section). Line 48. N2O as a source of stratospheric NOx was certainly known prior to Ravishankara's 2009 work, see McElroy, Khalil, Crutzen, etc. Lines 80-81. Rahn and Wahlen (1997) also contributed here with the first reference that describes a fractionation factor for stratospheric loss. Lines 82-84. Rahn et al. (1998) also contributed here with the first laboratory verification of wavelength dependent kinetic fractionation during photolysis. Line 116. "carbon isotopic composition of"??? Lines 141-142. For my own edification, doesn't this require dual bladders so that a specific depth range can be isolated? Lines 270-275. Mole fraction data from NEEM is substituted for with CSIRO/IUP/CIC/NOAA data. Does this induce a simple offset or a trending offset? In either case, by how much? Lines 289-299. The natural sources from land and sea have quite different production mechanisms as well as previously observed and predicted different isotopic signatures (Kim and Craig, 93; Rahn and Wahlen, 2000; and others) yet the isotopic model used here lumps all natural source into a single term. Given the detailed history that the authors are attempting to tease apart, this seems to me to be detrimental to their efforts. Please justify lumping the natural source into a single term. This should also be elaborated on further in the discussion section. Line 326. As well as Rahn and Wahlen 2000. Lines 429-439, discussion of inter-laboratory variability. The authors state that discrepancies do not exhibit a systematic shift and that the Sowers et al data had good agreement between two different laboratories but then they leave this conundrum hanging. Please elaborate a little on possibilities. Line 457. Regularization term is increased by how much? Lines 469-470. I find this paragraph confusing. The average annual emission of 3.5 TgN/yr in the last sentence should be the difference between 1940 and 2008 should it not? But this difference is 4.4 TgN/yr. I think I am confused because in the one case the natural term is included but in the other they are only considering the anthropogenic. In any case, this entire paragraph could be presented with more clarity. Lines 488-495 and lines 587-595. Observation of decadal variability in d15Navg and discussion of changes in relative contribution of sources

over time. If real, this is possibly the most important observation in the manuscript and needs to be dealt with in a much more considered manner. Intuitively, one would agree with their first statement, i.e. that the agricultural source would imprint the record more significantly in the earlier part of the record and decrease, in a relative sense, over time. This would mean that yes, the earliest human influences would be significantly depleted. As time goes on, fertilizer use becomes more controlled leading to less overuse and more limited flux of N2O accompanied by less isotopic discrimination. This, along with increased industrial production of N2O could hypothetically lead to the observed increase in d15Nanth over time (both avg and beta). This increase (seen in Fig. 4 right) peaks in the late 1980s however and proceeds to decrease significantly (~10 per mil for d15Navg and 20 per mil for d15Nbeta). This decrease is much more difficult to explain in a qualitative sense and in truth, is difficult to believe. One possibility is that industries are doing a better job of decreasing and/or capturing fugitive emissions which might increase the proportion of d15N depleted agriculture relative to industry but A) is there any evidence for this and B) would it yield this large of a result? A more detailed discussion of this is warranted given the subtly profound implications including discussion of potential artifacts in measuring/modeling that could also lead to the observed/modeled record. Line 524. Starting value of d18Oanth looks more like about 8 per mil to me but I'm looking at Fig. 4 because there is no Fig. 5. Lines 552-562. The authors perform a sensitivity study of changing the life time but it seems to me that there should also be sensitivity study of other terms, in particular F(exchange)which is a term that is poorly known. Also, given the two box model that is being used, it seems that a more appropriate lifetime would be stratospheric lifetime in conjunction with Xstrat given that this is the box where all of the N2O destruction takes place. Lines 563-573. This was also predicted by Rahn and Wahlen (2000), prior to any firn air measurements being made, where they predicted a -0.03 per mil/yr trend in 15Navg (identical to that on line 417) and a -0.03 per mil/yr trend in 18O (-0.02 per mil/yr on line 418 being within their estimated error). Lines 576-580. The "natural" component of the ocean source is estimated to be on the order of 4Tg N/yr. This new

"anthropogenic" component would then comprise a 25% increase in the ocean source. This gets back to my earlier comment on separating the natural source into land and ocean sources. Would this "new" oceanic N2O have an identical isotopic signature to the natural signature or would it be somehow different? In either case, it would certainly be distinct from the land signature. How would this be reflected in the temporal evolution of the firn records? General comments. On two occasions reference is made to Fig. 5 but no Fig. 5 exists. I assume they refer to Fig. 4? In the Appendix: Fig. A1 caption, left and right are switched. Figures C1 and C2 appear to be switched, Fig. 3 (page 45) precedes Fig. C2 (page 47) and there is a Fig. 3 and a Fig. C3 (or is it Fig. C3 and C4?). This is all rather sloppy. It is difficult for the reader to tease apart which data sets are new analyses and which were previously published. The new samples from NEEM are discussed thoroughly and the previously published data sets are referred to but nowhere is there an itemized tabulation of which data is associated which with specific publications and which, other than NEEM, if any, are new. In addition, there are two different records from NGRIP-01, one which is included in the analysis and one which is not but both are referred to with the same sample name. Please add a subscript or some other differentiating factor so that the reader does not have to try and sort this out for himself. Ultimately the authors conclude that "Based on the changes in the isotopes we conclude that the main contribution to N2O change in the atmosphere since 1940 is form soils, with agricultural soils being the principal anthropogenic component which is in line with previous studies." which is anticlimactic to say the least given the effort that went into sample collection, processing, analysis and modeling.
* * *

---

## Referee Comment (RC2) · Anonymous Referee #2 · 17 Nov 2016

Review for the following manuscript Journal: ACP Title: Constraining N2O emissions since 1940 by firn air isotope measurements in both hemispheres Author(s): M. Prokopiou et al. MS No.: acp-2016-487 MS Type: Research Article

The paper is generally well written. I appreciate the effort the authors put in in compiling all available firn air data and make the assessment. I don't have any serious major criticisms on their scientific approach and their interpretations but I do have major suggestions for their presentation. After the major comments are taken, I recommend for publication.

Major comments: 1. Box model calculation: The model parameters that kept invarying are not stated clearly. A table that lists all time independent parameters (crosstropopause exchange fluxes of isotopologues, natural fluxes, and their associated isotopic signatures, N2O lifetime, etc) will be helpful. In addition, a comparison with AR5 fluxes is useful. 2. Also box model: The derived time dependent variables. A table that summarizes the derived fluxes and isotopic values (average over a certain period) will be helpful, along with comparisons with other independent work by, for example, Park et al. and AR5. 3. What's the reason(s) behind for the elevated N2O flux in year 2008? 4. What's the reason(s) for the oscillating values in source/anthropogenic delta values in Figure 4? Moreover, if I understand correctly, natural N2Os are kept constant. I then expect to see the same time variability in anthropogenic as in source in Figure 4, but apparently the two are different. This highlights the usefulness of the major comment #1. 5. In addition to isotopic values, it will be useful and more informative to have isoflux for each process considered. A plot similar Figure 4 but for the respective flux (better also break into each process considered) is recommended.

Minor comments: 1. section 2.5: define all the variables used and no need to define variables not used. For example Fsink defined but not used. Fexch used but not defined. Also is epsilon_L the same as epsilon_app? Please check carefully the variables in the this section. 2. Line 445, additional decadal variability: raised also above in the major comment #4. What are the underlying mechanisms for the variability? Agricultural activity? Use of fertilizer? 3. Line 492, d15N^av: use the same notation throughout. In the figure, d15N is used. 4. Line 495, Fig 5: I believed you meant Fig. 4. Do the corrections for the remaining. 5. Table 3: Is your delta_atm,pi the same as Park et al.? If not, why not compare? If the same, then say it. 6. Same table, the last column double asterisk: what is it for? 7. Line 604, d15N_sp: not defined. You mentioned in line 36, but the term not defined. 8. d15N_sp is useful, please also show the time series in Fig 4.

---

## Author Comment (AC1) · 27 Jan 2017

RC Line 25: The 1940 atmospheric mole fraction, and hence the growth rate, differs from the determined by Battle et al. (1996) and to a lesser degree with Machida et al. 1995 please discuss (discussion section).

AR: Following the referee's suggestion we have added a discussion (see below) in the discussion section. "The N2O mole fraction atmospheric history from our multi-site reconstruction is in agreement with recent work from Meinshausen et al. (2016) who combined all available published N2O data (atmospheric, firn, ice) in order to reconstruct a historical atmospheric record of the past 2000 years. It differs slightly from the one determined by Battle et al. (1996) and to smaller extent with Machida

et al. (1995). Battle et al. (1996) collected firn air data and Machida et al. (1995) used ice data. Both studies used samples from a single Antarctic site. One could argue that the difference is due to an interhemispheric difference, but it is too large to be explained by this alone. In the past, N2O mole fraction measurements have been reported on different calibration scales, which is likely to explain part of the differences between individual studies. Furthermore, differences in the firn air model and possible differences between sites may contribute. In our case we used measurements from 5 sites to constrain our model while Battle et al. (1996) and Machida et al. (1995) used only one site. In addition, the atmospheric histories of up to 9 known gases (depending on site, Witrant et al. 2012) were used to constrain diffusivity in our model while Battle et al. (1996) only used two gases." Lines 549-563, Lines 738-740, Lines 811-813

RC Line 48: N2O as a source of stratospheric NOx was certainly known prior to Ravishankara's 2009 work see McElroy, Khalil, Crutzen, etc.

AR: The following references have been added to the manuscript: McElroy, M. B., and McConnell, J. C.: Nitrous Oxide: A natural source of stratospheric NO, Journal of Atmospheric Sciences, 28, 1095-1098, 1971. Crutzen, P. J.: The role of NO and NO2 in the chemistry of the troposphere and stratosphere, Annual review of earth and planetary sciences, 7, 443-472, 1979. Line 46, Lines 759-760, Lines 822-823

RC Lines 80-81: Rahn and Wahlen (1997) also contributed here with the first reference that describes a fractionation factor for stratospheric loss.

AR: The suggested reference has been added to the revised manuscript. Rahn, T., and Wahlen, M.: Stable isotope enrichment in stratospheric nitrous oxide, Science, 278, 1776-1778, doi: 10.1126/science.278.5344.1776, 1997. Line 79, Lines 865-866

RC Lines 82-84: Rahn et al. (1998) also contributed here with the first laboratory verification of wavelength dependent kinetic fractionation during photolysis.

AR: The suggested reference has been added to the revised manuscript. Rahn, T.,

Zhang, H., Wahlen, M., and Blake, G. A.: Stable isotope fractionation during ultraviolet photolysis of N2O, Geophys. Res. Lett., 25, 4489-4492, 1998. Line 83, Lines 867-868

RC Line 116: "Carbon composition of"?

AR: We corrected this omission and replaced it with "of its carbon composition". Line 115

RC Lines 141-142: For my own edification, doesn't this require dual bladders so that a specific depth range can be isolated?

AR: When firn samples are collected with this method, a new hole is drilled from the top. The hole is drilled to a certain depth and then the bladder is inserted down close to the bottom of the drill hole, so in practice the bottom of the drill hole plays the role of the second bladder that the referee indicated.

RC Lines 270-275: Mole fraction data from NEEM is substituted for, with CSIRO/IUP/CIC/NOAA data. Does this induce a simple offset or a trending offset? In either case by how much?

AR: NEEM data measured at IMAU was replaced by IUP/CIC/NOAA and CSIRO because IMAU data is less precise. The atmospheric trend reconstruction from IMAU-NEEM data, show in in black, in Fig. 1 has larger uncertainties than the trend scenario based on the more precise IUP/CIC/NOAA/CSIRO data (in red). The IMAU data based scenario is smoother because the model can reconstruct less details from more uncertain data. Thus the replacement of IMAU data with IUP/CIC/NOAA/CSIRO data does not induce an offset but leads to a more accurate and less smoothed output scenario. Lines 274-276, Lines 468-469, Lines 1029-1030, Lines 1169-1170, Lines 1191-1192 RC Lines 289-299: The natural sources from land and sea have quite different production mechanisms as well as previously observed and predicted different isotopic signatures (Kim and Craig, 1993; Rahn and Wahlen, 2000; and others) yet the isotopic model used here lumps all natural source into a single term. Given the detailed history

that the authors are attempting to tease apart, this seems to me detrimental to their efforts. Please justify keeping the natural source as a single term. This should also be elaborated on further in the discussion section.

AR: Yes, the isotope signatures of natural and terrestrial sources are different, and this has indeed been used to reconstruct contributions of marine and terrestrial sources to N2O variations in the past (Schilt et al., 2014). In the present study we focus on the change since pre-industrial times, where the strong increase in the N2O mole fraction suggests that this increase is dominated anthropogenic activities. Therefore we apply a model that assumes that the natural source strength has remained constant. This is clearly stated in the paper, but we have added some additional comments in the revised version. We would of course like to be able to differentiate between more processes, but our result indicates that with the present analytical precision, and given the uncertainties in source signatures, it is not really adequate to constrain additional degrees of freedom. In response to the referee comment we have added in the discussion that changes in natural sources that occur in parallel to the anthropogenic emissions cannot be distinguished with our approach, but may as well influence the results. Lines 333-338, Lines 595-597

RC Line 326: As well as Rahn and Wahlen 2000

AR: The suggested reference has been added to the revised manuscript. Rahn, T., and Wahlen, M.: A reassessment of the global isotopic budget of atmospheric nitrous oxide, Glob. Biogeochem. Cycl., 14, 537-543, 2000. Line 326, Lines 869-870

RC Lines 429-439: Discussion of interlaboratory variability. The authors state that discrepancies do not exhibit a systematic shift and that Sowers et al. data had good agreement between two different laboratories but then they leave this conundrum hanging. Please elaborate a little on possibilities.

AR: We realize that this is not fully satisfactory, but we have investigated this in quite some detail and cannot resolve the discrepancies A possible origin of the difference

could be based on the reconstruction model. Because the uncertainties on the South Pole data are large, compared to the other sites, the multi-site homogenization is more uncertain and less efficient (see Appendix A and C, Fig. A1 and C1-C3). Sampling uncertainty should also be taken into consideration since when pumping firn air and filling the sampling flasks you could encounter uncertainties (contamination, possible leak, fractionation, incomplete flask flushing etc). We have added this as additional discussion to the text. Lines 438-444

RC Line 457: Regularization term is increased by how much?

AR: The regularization factor was increased by a factor of 10. This was added in the revised manuscript. Lines 461-463, Line 633, Line 1109

RC Lines 469-470: I find this paragraph confusing. The average annual emission of 3.5 TgN/yr in the last sentence should be the difference between 1940-2008 should it not? But there the difference is 4.4 TgN/yr. I think I am confused because in the one case the natural term is included but in the other they are only considering the anthropogenic. In any case, this entire paragraph could be presented with more clarity.

AR: We have reworded this paragraph to remove the confusion. The average annual emission of 3.5 TgN/yr corresponds to the annual growth rate of 0.7 nmol mol–1 a-1 calculated between years 1995 and 2008. The total change the N2O mole fraction of (32±1) nmol mol–1 can be explained in the mass balance model by a (4.4±1.7) Tg a–1 N increase in the emissions from in 1940 to 2008. The paragraph has been updated to state the above clearly. Lines 474-476

RC Lines 488-495 and lines 587-595: Observation of decadal variability in d15Nav and discussion of changes in relative contribution of sources over time. If real, this is possibly the most important observation in the manuscript and needs to be dealt with much more considered manner. Intuitively, one would agree with their statement i.e. that the agricultural source would imprint the record more significantly in the earlier part of the record and decrease, in a relative sense, over time. This would mean that

yes, the earliest human influences would be significantly depleted. As time goes on, fertilizer use becomes more controlled leading to less overuse and more limited flux of N2O accompanied by less isotopic discrimination. This along with increased industrial production of N2O would hypothetically lead the observed increase in d15Nanth over time (both avg and beta). This increase (Fig. 4, right) peaks in the late 1980's however and proceeds to decrease significantly ($\sim$10% for d15Nav and $\sim$20% for d15Nb). This decrease is much more difficult to explain in a qualitative sense and in truth is difficult to believe. One possibility is that industries are a doing a better job of decreasing and/or capturing fugitive emissions which might increase in the proportion of d15N depleted agriculture relative to industry but: A. Is there evidence of this? B. Would it yield this large of a result? A more detailed discussion of this is warranted given the subtly profound implications including discussion of potential artifacts in measuring and modeling that could also lead the observed modeled record.

AR: We thank the referee for this a comment. We actually tried to keep a balance between discussing possible scenarios (first part of the referee comment) and examine whether the reconstructed changes are realistic (second part of the referee comment). Independent quantitative evidence for some of the suggested changes is actually available from inventory information, which was not included in the original version of our manuscript. According to FAO statistics (http://www.fao.org/faostat/en/#data/GY/visualize), emissions from synthetic nitrogenous fertilizers increased between 1961 and 1985, then stayed relatively constant or even decreased until 2000, and increased again after 2000. The reasons of the decrease between 1985 and 2000 are a small turn to organic soil cultivation in combination with more efficient agricultural methods and fertilizer use. This qualitatively matches the temporal evolutions of our reconstructed source signatures, but as the referee points out the observed isotopic signature change after 1985 is quite large, especially for $\delta$15Nav and $\delta$15N$\beta$ . This is why we did not discuss this in more detail in the original manuscript. In the revised version, we added this information, but also point out that this effect is likely not strong enough to explain the reconstructed isotope signal.

In the manuscript, we do discuss the fact that (part of) decadal variability may originate from small undulations on the reconstructed scenarios, since the emissions are related to the derivative of the trend. It is possible to draw straight lines within the uncertainty envelopes of the scenarios, and therefore the reconstructed decadal variability may not be robust but a product of the reconstruction procedure. Indeed, increasing the regularization term by a factor of 10 leads to much smoother (nearly straight) lines that fit inside the uncertainty envelopes. Therefore the decadal variability may not be realistic, and we do not want to put too much emphasis on the interpretation in terms of underlying processes. We realize that this is a bit unsatisfactory, but given the large uncertainties we think that the level of interpretation that we give is adequate. We do not want to put forward scenarios that are unrealistic and not really backed up by the data. Lines 621-637

RC Line 524: Starting value of d18Oanth, looks like about 8% to me but I am looking at Fig. 4 because there is no Fig. 5.

AR: The mistake has been corrected and substituted with (7.7±2.6) ‰ Line 529

RC Lines 552-562: The authors perform a sensitivity study of changing the lifetime but it seems to me that there should also be a sensitivity study on other terms, in particular F (exchange) which is a term that is poorly known. Also given the two box model that is being used it seems that more appropriate lifetime would be stratospheric lifetime in conjunction with Xstrat given that this is the box where all N2O destruction takes place.

AR: Sensitivity tests on the magnitude of Fexch have been added in the Appendix D. The results show that when the Fexch value is low, then less N2O is returned to the troposphere, contrary, when Fexch is high more N2O is returned. The study showed that Fexch has little effect on the isotopic signature results, thus we concluded that only the flux is sensitive to the choice of Fexch value while the isotopic composition is not. The use of global mean lifetime is correct because eq. 3 in the manuscript refers to the total atmospheric burden and not the stratospheric burden. The mean stratospheric

lifetime would be about 10 times smaller than the global mean lifetime. Lines 582-586, Lines 1148-1166, Lines 1190-1213

RC Lines 563-573: This was also predicted by Rahn and Wahlen (2000), prior to any firn air measurements being made, where they predicted a -0.03 permil/yr trend in 15Nav (identical to that on the line 417) and a -0.03 prmil/yr trend in 18O (-0.02 permil/yr on line 418 being within the estimated error).

AR: This has been included in the revised manuscript. Lines 593-594, Lines 869-870

RC Lines 576-580: The 'natural' component of the ocean source is estimated to be on the order of 4Tg N/yr. This new 'anthropogenic' component would then comprise a 25% increase in the ocean source. This gets back to my earlier comment on separating the natural source into land and ocean sources. Would this 'new' oceanic N2O have an identical isotopic signature to the natural signature or would it be somehow different? In either case, it would certainly be distinct from the land signature. How would this be reflected in the temporal evolution of the firn records?

AR: As mentioned above, we cannot really constrain more free parameters, and we have chosen to lump all parts of the "anthropogenic" source together. Here we discuss that variations in different components of the anthropogenic source may leave temporal signals in the source signature. Snider et al. (2015) made a meta-analysis of previously published source signature studies and concluded that freshwater bulk isotope signatures are (-7.78±9.72) ‰ and (40.75±9.63) ‰ for $\delta$15Nav and $\delta$18O respectively. Similarly for marine waters the results were (5.14±1.93) for $\delta$15Nav and (44.76±3.62) for $\delta$18O. We feel that it is not possible at present to make a quantitative statement, given the available information both from bottom-up studies and isotope source signature studies, and therefore discuss these effects qualitatively only.

General comments: RC: On two occasions reference is made to Fig. 5, but no Fig. 5 exists. I assume they refer to Fig. 4? In the Appendix: Fig. A1 caption, left and right are switched. Figures CI and C2 appear to be switched, Fig. 3 (page 45) precedes

[Figure]

Fig. C2 (page 47) and there is a Fig. 3 and a Fig. C3 (or is it Fig. C3 and C4?). This is all rather sloppy. It is difficult for the reader to tease apart which data sets are new analyses and which were previously published.

AR: We apologize for the mislabeling, and these errors were corrected in the revised version.

RC: The new samples from NEEM are discussed thoroughly and the previously published data sets are referred to but nowhere is there an itemized tabulation of which data is associated which with specific publications and which, other than NEEM, if any, are new.

AR: This information has been added in the revised manuscript in the revised Table 1. Lines 953-960

RC: In addition, there are two different records from NGRIP-01, one which is included in the analysis and one which is not but both are referred to with the same sample name. Please add a subscript or some other differentiating factor so that the reader does not have to try and sort this out for himself.

AR: The requested information has been added in the revised manuscript. A subscript indicating the differentiation between the two publications is used (NGRIP-01Ishijima, NGRIP-01Bernard) throughout the manuscript. Line 211, Line 213, Line 393, Lines 428-429, Line 954, Line 1010, Lines 1055-1056, Line 1102

RC: Ultimately the authors conclude that 'Based on the changes in the isotopes we conclude that the main contribution to N2O change in the atmosphere since 1940 is from soils, with agricultural soils being the principal anthropogenic component which is in line with previous studies'. Which is anticlimactic to say the least given the effort that went into sample collection, processing, analysis and modeling.

AR: We agree that this part of the conclusion should be modified. We set out with this project to detect possible temporal changes in the isotopic composition, but we

find that such changes are not clearly quantifiable with the present analytical precision. Therefore the conclusion is a bit negative (as presented in the abstract), but have described our results and the limitations more quantitatively in the revised version.

[Figure]

[Figure]

**Fig. 1.** Figure 1: Firn air trend reconstruction using only NEEM measurements from IMAU laboratory (black line) with uncertainty envelopes (dashed black lines) compared to firn air trend reconstruction using N

[Figure]

[Figure]

**Fig. 2.** Figure 2: Global synthetic fertiliser emissions from 1961-2014 in CO2eq. Figure retrieved from FAO statistics (http://www.fao.org/faostat/en/#data/GY/visualize)

---

## Author Comment (AC2) · 27 Jan 2017

Major comments: RC 1. Box model calculation: The model parameters that kept invarying are not stated clearly. A table that list all time independent parameteres (cross-tropopause exchange fluxes of isotopologues, natural fluxes and their associated isotopic signatures, N2O lifetime, etc) will be helpful. In addition, a comparison with AR5 fluxes is useful. RC 2. Also box model: the derived time dependent variables. A table that summarizes the derived fluxes and isotopic values (average over a certain period) will be helpful, along with comparisons with other independent work by, for example, Park et al. and AR5.

Author's response to major comments 1 and 2: We realise that a more detailed presentation of the parameters used is needed therefore we have substituted Table 3 where only natural and anthropogenic isotopic signature results were presented with a more detailed version including stratospheric loss fluxes and isotopic signatures, N2O lifetime, natural and anthropogenic fluxes as in the two-box model. The values were compared to Park et al. (2012) because they provide results not only for fluxes but also for isotopic signatures. We did not include a comparison with the AR5 for the reason that it provides us only with flux results not isotopic signature ones. Lines 989-995

RC 3. What's the reason(s) behind for the elevated N2O flux in year 2008?

AR: We suspect the referee refers to the very slightly increasing emission strength at the end of the reconstructed record. This apparent upwards trend is likely not significant for our construction and we have not discussed it in more detail. We shortly stated this in the revised manuscript. Lines 478-480

RC 4. What's the reason(s) for the oscillating values in source/anthropogenic delta values in Fig. 4? Moreover, if I understand correctly, natural N2Os are kept constant. I then expect to see the same time variability in anthropogenic as in source in Figure 4, but apparently the two are different. This highlights the usefulness of the major comment #1.

AR: The reason why the oscillations of the total and the anthropogenic source are not the same is that in our mass balance model the total source is regarded as the sum of a constant natural source and a changing anthropogenic source, which was small in the beginning of the record and larger at the end of the record. Therefore, changes in the total source signature in the beginning of the record require a substantially stronger isotope signal in the (small) anthropogenic source at that time compared to the (large) anthropogenic source at the end of the record. This was also stated in the manuscript. To make this more comprehensive we have added in Fig. 3 (bottom panel) the assumed constant, natural source, also. Lines 515-520, Lines 1019-1037

RC 5. In addition to isotopic values, it will be useful and more informative to have isoflux

for each process considered. A plot similar Fig. 4 but for the respective flux (better also break into each process considered is recommended).

AR: We have considered adding isofluxes to the manuscript, but since we only distinguish between a natural and an anthropogenic source this does not seem to add very useful information in our opinion. If – as the referee suggested – we were able to distinguish different processes it would indeed be useful, but since we cannot do that, we prefer not to add a discussion on isofluxes.

Minor comments: RC 1. section 2.5: define all the variables used and no need to define variables not used. For example Fsink defined but not used. Fexch used but not defined. Also is epsilon_L the same as epsilon_app? Please check carefully the variables in this section.

AR: The section has been updated, Fsink is replaced by L, epsilon_L is the same as epsilon_app and we have adopted only the first and Fexch is defined in Table 3. Lines 295-296, Line 314, Lines 317-318, Line 323, Line 324, Line 325, Line 328, Line 672, Line 675, Lines 989-996

RC 2. Line 445: additional decadal variability: raised also above in the major comment #4. What are the underlying mechanisms for the variability? Agricultural activity? Use of fertilizer?

AR: Yes these are the mechanisms we describe and we added some more clarification in the discussion section. Lines 621-637

RC 3. Line 492: d15Nav" is the same notation throughout, in the figure d15N is used.

AR: The notation d15N in the figure was replaced with d15Nav.

RC 4. Line 495: Fig.5, I believed you meant Fig. 4. Do the corrections for the remaining.

AR: Thank you for pointing this out, it has been corrected.

RC 5. Table 3: Is your delta_atm,pi the same as Park et al.? If not, why not compare? If the same then say it.

AR: The delta_atm,pi is the same as Park et al. and it is mention in the footnote denoted with an asterisk located below table 3. Lines 992-995

RC 6. Same table, the last column double asterisk: what is it for?

AR:Thanks for noting this, the double asterisks was removed.

RC 7. Line 604: d15N_sp: not defined. You mentioned in line 36, but the term not defined.

AR:d15N_sp is now defined in line 37.

RC 8. d15N_sp is useful: please also show the time series in Fig. 4

AR: The information has been added in the revised manuscript. Lines 1038-1045
* * *

---

## Author Response (AR1)

**Reply to referee Thomas Rahn comments**

*Ref. Com. Line 25: The 1940 atmospheric mole fraction, and hence the growth rate, differs from the determined by Battle et al. (1996) and to a lesser degree with Machida et al. 1995 please discuss (discussion section).*

**Author's Resp.:** Following the referee's suggestion we have added a discussion (see below) in the discussion section.
"The $N_2O$ mole fraction atmospheric history from our multi-site reconstruction is in agreement with recent work from Meinshausen et al. (2016) who combined all available published $N_2O$ data (atmospheric, firn, ice) in order to reconstruct a historical atmospheric record of the past 2000 years. It differs slightly from the one determined by Battle et al. (1996) and to smaller extent with Machida et al. (1995).

Battle et al. (1996) collected firn air data and Machida et al. (1995) used ice data. Both studies used samples from a single Antarctic site. One could argue that the difference is due to an interhemispheric difference, but it is too large to be explained by this alone. In the past, N2O mole fraction measurements have been reported on different calibration scales, which is likely to explain part of the differences between individual studies. Furthermore, differences in the firn air model and possible differences between sites may contribute. In our case we used measurements from 5 sites to constrain our model while Battle et al. (1996) and Machida et al. (1995) used only one site. In addition, the atmospheric histories of up to 9 known gases (depending on site, Witrant et al. 2012) were used to constrain diffusivity in our model while Battle et al. (1996) only used two gases."
**Author's changes:** Lines 549-563, Lines 738-740, Lines 811-813

*Ref. Com. Line 48: $N_2O$ as a source of stratospheric NOx was certainly known prior to Ravishankara's 2009 work see McElroy, Khalil, Crutzen, etc.*

**Author's Resp.:** The following references have been added to the manuscript:
McElroy, M. B., and McConnell, J. C.: Nitrous Oxide: A natural source of stratospheric NO, Journal of Atmospheric Sciences, 28, 1095-1098, 1971.

Crutzen, P. J.: The role of NO and $NO_2$ in the chemistry of the troposphere and stratosphere, Annual review of earth and planetary sciences, 7, 443-472, 1979.
**Author's changes:** Line 46, Lines 760-761, Lines 822-823

*Ref. Com. Lines 80-81: Rahn and Wahlen (1997) also contributed here with the first reference that describes a fractionation factor for stratospheric loss.*

**Author's Resp.:** The suggested reference has been added to the revised manuscript.
Rahn, T., and Wahlen, M.: Stable isotope enrichment in stratospheric nitrous oxide, Science, 278, 1776-1778, doi: 10.1126/science.278.5344.1776, 1997.
**Author's changes:** Line 79, Lines 865-866

*Ref. Com. Lines 82-84: Rahn et al. (1998) also contributed here with the first laboratory verification of wavelength dependent kinetic fractionation during photolysis.*

**Author's Resp.:** The suggested reference has been added to the revised manuscript. Rahn, T., Zhang, H., Wahlen, M., and Blake, G. A.: Stable isotope fractionation during ultraviolet photolysis of $N_2O$, Geophys. Res. Lett., 25, 4489-4492, 1998.
**Author's changes:** Line 83, Lines 867-868

*Ref. Com. Line 116: "Carbon composition of"?*

**Author's Resp.:** We corrected this omission and replaced it with "of its carbon composition".
**Author's changes:** Line 115

*Ref. Com. Lines 141-142: For my own edification, doesn't this require dual bladders so that a specific depth range can be isolated?*

**Author's Resp.:** When firn samples are collected with this method, a new hole is drilled from the top. The hole is drilled to a certain depth and then the bladder is inserted down close to the bottom of the drill hole, so in practice the bottom of the drill hole plays the role of the second bladder that the referee indicated.

*Ref. Com. Lines 270-275: Mole fraction data from NEEM is substituted for, with CSIRO/IUP/CIC/NOAA data. Does this induce a simple offset or a trending offset? In either case by how much?*

**Author's Resp.:** NEEM data measured at IMAU was replaced by IUP/CIC/NOAA and CSIRO because IMAU data is less precise. The atmospheric trend reconstruction from IMAU-NEEM data, show in in black, in Fig. 1 has larger uncertainties than the trend scenario based on the more precise IUP/CIC/NOAA/CSIRO data (in red). The IMAU data based scenario is smoother because the model can reconstruct less details from more uncertain data. Thus the replacement of IMAU data with IUP/CIC/NOAA/CSIRO data does not induce an offset but leads to a more accurate and less smoothed output scenario.
**Author's changes:** Lines 274-276, Lines 468-469, Lines 1018-1019, Lines 1029-1030, Lines 1160-1161, Lines 1183-1184

[Figure]

Figure 1: Firn air trend reconstruction using only NEEM measurements from IMAU laboratory (black line) with uncertainty envelopes (dashed black lines) compared to firn air trend reconstruction using NEEM measurements from IUP/CIC/NOAA and CSIRO atmospheric (red line) with corresponding uncertainty envelopes (dashed red lines).

***Ref. Com. Lines 289-299:*** *The natural sources from land and sea have quite different production mechanisms as well as previously observed and predicted different isotopic signatures (Kim and Craig, 1993; Rahn and Wahlen, 2000; and others) yet the isotopic model used here lumps all natural source into a single term. Given the detailed history that the authors are attempting to tease apart, this seems to me detrimental to their efforts. Please justify keeping the natural source as a single term. This should also be elaborated on further in the discussion section.*

**Author's Resp.:** Yes, the isotope signatures of natural and terrestrial sources are different, and this has indeed been used to reconstruct contributions of marine and terrestrial sources to $N_2O$ variations in the past (Schilt et al., 2014). In the present study we focus on the change since pre-industrial times, where the strong increase in the $N_2O$ mole fraction suggests that this increase is dominated anthropogenic activities. Therefore we apply a model that assumes that the natural source strength has remained constant. This is clearly stated in the paper, but we have added some additional comments in the revised version. We would of course like to be able to differentiate between more processes, but our result indicates that with the present analytical precision, and given the uncertainties in source signatures, it is not really adequate to constrain additional degrees of freedom. In response to the referee comment we have added in the discussion that changes in natural sources that occur in parallel to the anthropogenic emissions cannot be distinguished with our approach, but may as well influence the results.
**Author's changes:** Lines 333-338, Lines 598-599

***Ref. Com. Line 326:*** *As well as Rahn and Wahlen 2000*

**Author's Resp.:** The suggested reference has been added to the revised manuscript. Rahn, T., and Wahlen, M.: A reassessment of the global isotopic budget of atmospheric nitrous oxide, Glob. Biogeochem. Cycl., 14, 537-543, 2000.
**Author's changes:** Line 326, Lines 869-870

*Ref. Com. Lines 429-439: Discussion of interlaboratory variability. The authors state that discrepancies do not exhibit a systematic shift and that Sowers et al. data had good agreement between two different laboratories but then they leave this conundrum hanging. Please elaborate a little on possibilities.*

**Author's Resp.:** We realize that this is not fully satisfactory, but we have investigated this in quite some detail and cannot resolve the discrepancies A possible origin of the difference could be based on the reconstruction model. Because the uncertainties on the South Pole data are large, compared to the other sites, the multi-site homogenization is more uncertain and less efficient (see Appendix A and C, Fig. A1 and C1-C3). Sampling uncertainty should also be taken into consideration since when pumping firn air and filling the sampling flasks you could encounter uncertainties (contamination, possible leak, fractionation, incomplete flask flushing etc). We have added this as additional discussion to the text.
**Author's changes:** Lines 438-444

*Ref. Com. Line 457: Regularization term is increased by how much?*

**Author's Resp.:** The regularization factor was increased by a factor of 10. This was added in the revised manuscript.
**Author's changes:** Lines 461-463, Line 635, Line 1098

*Ref. Com. Lines 469-470: I find this paragraph confusing. The average annual emission of 3.5 TgN/yr in the last sentence should be the difference between 1940-2008 should it not? But there the difference is 4.4 TgN/yr. I think I am confused because in the one case the natural term is included but in the other they are only considering the anthropogenic. In any case, this entire paragraph could be presented with more clarity.*

**Author's Resp.:** We have reworded this paragraph to remove the confusion. The average annual emission of 3.5 TgN/yr corresponds to the annual growth rate of 0.7 nmol mol$^{-1}$ a$^{-1}$ calculated between years 1995 and 2008. The total change the $N_2O$ mole fraction of $(32\pm1)$ nmol mol$^{-1}$ can be explained in the mass balance model by a $(4.4\pm1.7)$ Tg a$^{-1}$ N increase in the emissions from in 1940 to 2008. The paragraph has been updated to state the above clearly.
**Author's changes:** Lines 474-476

*Ref. Com. Lines 488-495 and lines 587-595: Observation of decadal variability in d15Nav and discussion of changes in relative contribution of sources over time. If real, this is possibly the most important observation in the manuscript and needs to be dealt with much more considered manner. Intuitively, one would agree with their statement i.e. that the agricultural source would imprint the record more significantly in the earlier part of the record and decrease, in a relative sense, over time. This would mean that yes, the earliest human influences would be significantly depleted.*

*As time goes on, fertilizer use becomes more controlled leading to less overuse and more limited flux of N2O accompanied by less isotopic discrimination. This along with increased industrial production of N2O would hypothetically lead the observed increase in d15Nanth over time (both avg and beta). This increase (Fig. 4, right) peaks in the late 1980's however and proceeds to decrease significantly (~10% for d15Nav and ~20% for d15Nb). This decrease is much more difficult to explain in a qualitative sense and in truth is difficult to believe.*

*One possibility is that industries are a doing a better job of decreasing and/or capturing fugitive emissions which might increase in the proportion of d15N depleted agriculture relative to industry but:*

*A. Is there evidence of this?*

*B. Would it yield this large of a result?*

*A more detailed discussion of this is warranted given the subtly profound implications including discussion of potential artifacts in measuring and modeling that could also lead the observed modeled record.*

**Author's Resp.:** We thank the referee for this a comment. We actually tried to keep a balance between discussing possible scenarios (first part of the referee comment) and examine whether the reconstructed changes are realistic (second part of the referee comment). Independent quantitative evidence for some of the suggested changes is actually available from inventory information, which was not included in the original version of our manuscript. According to FAO statistics (http://www.fao.org/faostat/en/#data/GY/visualize), emissions from synthetic nitrogenous fertilizers increased between 1961 and 1985, then stayed relatively constant or even decreased until 2000, and increased again after 2000. The reasons of the decrease between 1985 and 2000 are a small turn to organic soil cultivation in combination with more efficient agricultural methods and fertilizer use. This qualitatively matches the temporal evolutions of our reconstructed source signatures, but as the referee points out the observed isotopic signature change after 1985 is quite large, especially for $\delta^{15}N^{av}$ and $\delta^{15}N^{\beta}$ . This is why we did not discuss this in more detail in the original manuscript. In the revised version, we added this information, but also point out that this effect is likely not strong enough to explain the reconstructed isotope signal.

In the manuscript, we do discuss the fact that (part of) decadal variability may originate from small undulations on the reconstructed scenarios, since the emissions are related to the derivative of the trend. It is possible to draw straight lines within the uncertainty envelopes of the scenarios, and therefore the reconstructed decadal variability may not be robust but a product of the reconstruction procedure. Indeed, increasing the regularization term by a factor of 10 leads to much smoother (nearly straight) lines that fit inside the uncertainty envelopes. Therefore the decadal variability may not be realistic, and we do not want to put too much emphasis on the interpretation in terms of underlying processes. We realize that this is a bit unsatisfactory, but given the large uncertainties we think that the level of interpretation that we give is adequate. We do not want to put forward scenarios that are unrealistic and not really backed up by the data.

**Author's changes:** Lines 621-637

[Figure]

**Emissions (CO2 equivalent), Synthetic Nitrogen fertilizers**

- 2014

*Ref. Com. Line 524:* *Starting value of d18Oanth, looks like about 8% to me but I am looking at Fig. 4 because there is no Fig. 5.*

**Author's Resp.:** The mistake has been corrected and substituted with (7.7±2.6) ‰.
**Author's changes:** Line 529

*Ref. Com. Lines 552-562:* *The authors perform a sensitivity study of changing the lifetime but it seems to me that there should also be a sensitivity study on other terms, in particular F (exchange) which is a term that is poorly known. Also given the two box model that is being used it seems that more appropriate lifetime would be stratospheric lifetime in conjunction with Xstrat given that this is the box where all N2O destruction takes place.*

**Author's Resp.:** Sensitivity tests on the magnitude of Fexch have been added in the Appendix D. The results show that when the Fexch value is low, then less $N_2O$ is returned to the troposphere, contrary, when Fexch is high more $N_2O$ is returned. The study showed that Fexch has little effect on the isotopic signature results, thus we concluded that only the flux is sensitive to the choice of Fexch value while the isotopic composition is not.
The use of global mean lifetime is correct because eq. 3 in the manuscript refers to the total atmospheric burden and not the stratospheric burden. The mean stratospheric lifetime would be about 10 times smaller than the global mean lifetime.
**Author's changes:** Lines 582-586, Lines 1117-1118, Lines 1124-1126, Lines 1140-1158, Lines 1177-1205

*Ref. Com. Lines 563-573:* *This was also predicted by Rahn and Wahlen (2000), prior to any firn air measurements being made, where they predicted a -0.03 permil/yr trend in 15Nav (identical to that on the line 417) and a -0.03 prmil/yr trend in 18O (-0.02 permil/yr on line 418 being within the estimated error).*

**Author's Resp.:** This has been included in the revised manuscript.

**Author's changes:** Lines 595-596, Lines 869-870

***Ref. Com. Lines 576-580:*** *The 'natural' component of the ocean source is estimated to be on the order of 4Tg N/yr. This new 'anthropogenic' component would then comprise a 25% increase in the ocean source. This gets back to my earlier comment on separating the natural source into land and ocean sources. Would this 'new' oceanic N2O have an identical isotopic signature to the natural signature or would it be somehow different? In either case, it would certainly be distinct from the land signature. How would this be reflected in the temporal evolution of the firn records?*

**Author Resp.:** As mentioned above, we cannot really constrain more free parameters, and we have chosen to lump all parts of the "anthropogenic" source together. Here we discuss that variations in different components of the anthropogenic source may leave temporal signals in the source signature. Snider et al. (2015) made a meta-analysis of previously published source signature studies and concluded that freshwater bulk isotope signatures are (-7.78±9.72) ‰ and (40.75±9.63) ‰ for $\delta^{15}N^{av}$ and $\delta^{18}O$ respectively. Similarly for marine waters the results were (5.14±1.93) for $\delta^{15}N^{av}$ and (44.76±3.62) for $\delta^{18}O$. We feel that it is not possible at present to make a quantitative statement, given the available information both from bottom-up studies and isotope source signature studies, and therefore discuss these effects qualitatively only.

**General comments:**
***Ref. Com.****: On two occasions reference is made to Fig. 5, but no Fig. 5 exists. I assume they refer to Fig. 4? In the Appendix: Fig. A1 caption, left and right are switched. Figures CI and C2 appear to be switched, Fig. 3 (page 45) precedes Fig. C2 (page 47) and there is a Fig. 3 and a Fig. C3 (or is it Fig. C3 and C4?). This is all rather sloppy. It is difficult for the reader to tease apart which data sets are new analyses and which were previously published.*

**Author's Resp.:** We apologize for the mislabeling, and these errors were corrected in the revised version.
**Author's changes:** Line 485, Line 501, Line 517, Line 530, Line 589, Line 1044, Line 1046, Lines 1088-1092, Lines 1093-1096

***Ref. Com.****: The new samples from NEEM are discussed thoroughly and the previously published data sets are referred to but nowhere is there an itemized tabulation of which data is associated which with specific publications and which, other than NEEM, if any, are new.*

**Author's Resp.:** This information has been added in the revised manuscript in the revised Table 1.
**Author's changes:** Lines 953-960

***Ref. Com.****: In addition, there are two different records from NGRIP-01, one which is included in the analysis and one which is not but both are referred to with the same sample name. Please add a subscript or some other differentiating factor so that the reader does not have to try and sort this out for himself.*

**Author Resp.:** The requested information has been added in the revised manuscript. A subscript indicating the differentiation between the two publications is used (NGRIP-01$_{Ishijima}$, NGRIP-01$_{Bernard}$) throughout the manuscript.
**Author's changes:** Line 211, Line 213, Line 393, Lines 428-429, Line 954, Line 999, Line 1013, Line 1015, Line 1044, Line 1045, Lines 1080-1087, Line 1091

*Ref. Com.: Ultimately the authors conclude that 'Based on the changes in the isotopes we conclude that the main contribution to N2O change in the atmosphere since 1940 is from soils, with agricultural soils being the principal anthropogenic component which is in line with previous studies'. Which is anticlimactic to say the least given the effort that went into sample collection, processing, analysis and modeling.*

**Author Resp.:** We agree that this part of the conclusion should be modified. We set out with this project to detect possible temporal changes in the isotopic composition, but we find that such changes are not clearly quantifiable with the present analytical precision. Therefore the conclusion is a bit negative (as presented in the abstract), but have described our results and the limitations more quantitatively in the revised version.

**Reply to Anonymous referee comments**

**Major comments:**
*Ref. Com. 1: Box model calculation: The model parameters that kept invarying are not stated clearly. A table that list all time independent parameteres (cross-tropopause exchange fluxes of isotopologues, natural fluxes and their associated isotopic signatures, N2O lifetime, etc) will be helpful. In addition, a comparison with AR5 fluxes is useful.*
*Ref. Com. 2: Also box model: the derived time dependent variables. A table that summarizes the derived fluxes and isotopic values (average over a certain period) will be helpful, along with comparisons with other independent work by, for example, Park et al. and AR5.*
**Author's response to major comments 1 and 2:**
We realise that a more detailed presentation of the parameters used is needed therefore we have substituted Table 3 where only natural and anthropogenic isotopic signature results were presented with a more detailed version including stratospheric loss fluxes and isotopic signatures, N$_2$O lifetime, natural and anthropogenic fluxes as in the two-box model. The values were compared to Park et al. (2012) because they provide results not only for fluxes but also for isotopic signatures. We did not include a comparison with the AR5 for the reason that it provides us only with flux results not isotopic signature ones.
**Author's changes:** Lines 989-995

*Ref. Com. 3: What's the reason(s) behind for the elevated N2O flux in year 2008?*

**Author's Resp.:** We suspect the referee refers to the very slightly increasing emission strength at the end of the reconstructed record. This apparent upwards trend is likely not significant for our construction and we have not discussed it in more detail. We shortly stated this in the revised manuscript.
**Author's changes:** Lines 478-480

***Ref. Com*.4:** *What's the reason(s) for the oscillating values in source/anthropogenic delta values in Fig. 4? Moreover, if I understand correctly, natural N2Os are kept constant. I then expect to see the same time variability in anthropogenic as in source in Figure 4, but apparently the two are different. This highlights the usefulness of the major comment #1.*

**Author's Resp.:** The reason why the oscillations of the total and the anthropogenic source are not the same is that in our mass balance model the total source is regarded as the sum of a constant natural source and a changing anthropogenic source, which was small in the beginning of the record and larger at the end of the record. Therefore, changes in the total source signature in the beginning of the record require a substantially stronger isotope signal in the (small) anthropogenic source at that time compared to the (large) anthropogenic source at the end of the record. This was also stated in the manuscript. To make this more comprehensive we have added in Fig. 3 (bottom panel) the assumed constant, natural source, also.
**Author's changes:** Lines 515-520, Lines 1019-1037

***Ref. Com. 5:*** *In addition to isotopic values, it will be useful and more informative to have isoflux for each process considered. A plot similar Fig. 4 but for the respective flux (better also break into each process considered is recommended).*

**Author's Resp.:** We have considered adding isofluxes to the manuscript, but since we only distinguish between a natural and an anthropogenic source this does not seem to add very useful information in our opinion. If – as the referee suggested – we were able to distinguish different processes it would indeed be useful, but since we cannot do that, we prefer not to add a discussion on isofluxes.

**Minor comments:**
***Ref. Com. 1 section 2.5:*** *define all the variables used and no need to define variables not used. For example Fsink defined but not used. Fexch used but not defined. Also is epsilon_L the same as epsilon_app? Please check carefully the variables in this section.*

**Author's Resp.:** The section has been updated, Fsink is replaced by L, epsilon_L is not the same as epsilon_app. Epsilon_L is constrained by epsilon_app but the numerical values differ depending on F_exch and the lifetime. Fexch is defined in Table 3.
**Author's changes:** Lines 295-296, Lines 317-318, Line 320-321, Lines 989-995

***Ref. Com. 2:*** *Line 445: additional decadal variability: raised also above in the major comment #4. What are the underlying mechanisms for the variability? Agricultural activity? Use of fertilizer?*

**Author's Resp.:** Yes these are the mechanisms we describe and we added some more clarification in the discussion section.
**Author's changes:** Lines 621-637

***Ref. Com. 3***: *Line 492: d15Nav" is the same notation throughout, in the figure d15N is used.*

**Author'sResp.:** The notation d15N in the figure was replaced with d15Nav.
**Author's changes:** Line 26, Line 32, Line 69, Line 70, Line 164, Line 166, Line 167, Line 170, Line 216, Line 224, Line 238, Line 247, Line 315, Line 414, Line 416, Line 430, Line 450, Line 494, Line 498, Line 501, Line 506, Line 520, Line 521, Line 522, Line 527, Line 537, Line 538, Line 544, Line 607, Line 629, Line 640, Line 697, Line 703, Line 708, Line 972-973, Line 991-995, Line 1029, Line 1058, Line 1134, Line 1136, Line 1172, Line 1197

*Ref. Com. 4: Line 495: Fig.5, I believed you meant Fig. 4. Do the corrections for the remaining.*

**Author Resp.:** Thank you for pointing this out, it has been corrected.

*Ref. Com. 5: Table 3: Is your delta_atm,pi the same as Park et al.? If not, why not compare? If the same then say it.*

**Author's Resp.:** The delta_atm,pi is the same as Park et al. and it is mention in the footnote denoted with an asterisk located below table 3.
**Author's changes:** Lines 992-995

*Ref. Com. 6: Same table, the last column double asterisk: what is it for?*

**Author's Resp.:** Thanks for noting this, the double asterisks was removed.

*Ref. Com. 7: Line 604: d15N_sp: not defined. You mentioned in line 36, but the term not defined.*

**Author's Resp.:** d15N_sp is now defined in line 37.
**Author's changes:** Line 37

*Ref. Com. 8: d15N_sp is useful: please also show the time series in Fig. 4*

**Author's Resp.:** The information has been added in the revised manuscript.
**Author's changes:** Lines 1029-1034

[revised manuscript text omitted]